# MULTIMODAL BANKING DATASET: UNDERSTANDING CLIENT NEEDS THROUGH EVENT SEQUENCES

## ABSTRACT

Financial organizations collect a huge amount of data about clients that typically has a temporal (sequential) structure and is collected from multiple sources (modalities). However, despite the urgent practical need, developing deep learning techniques suitable to handle such data is limited by the absence of large open-source multi-source real-world datasets of event sequences. To fill this gap mainly caused by security reasons, we present the industrial-scale publicly available multimodal banking dataset, MBD, that contains more than 2M corporate clients with several data sources: 950M bank transactions, 1B geo position events, 5M embeddings of dialogues with technical support and monthly aggregated purchases of four bank's products. All entries are properly anonymized from real proprietary bank data. Moreover, we introduce a novel multimodal benchmark incorporating our MBD and two open-source financial datasets. We provide numerical results demonstrating the superiority of fusion baselines over single-modal techniques for each task. Moreover, our anonymization techniques still save all significant information for introduced downstream tasks.

**Code Link**: https://anonymous.4open.science/r/MBD-034B/
**Dataset Link**: https://disk.yandex.ru/d/Pk9Mhx70VnUzbA

## 1 INTRODUCTION

The key tasks in the banking industry, such as campaigning, fraud detection, credit risk assessment, customer segmentation, and personalized recommendations, heavily rely on various aspects of clients' financial activities, e.g., product purchase history. This data, spanning extended periods, is typically annotated with temporal information, forming what is known as *event sequences* Babaev et al. (2022); Udovichenko et al. (2024); Yeshchenko & Mendling (2022); Kolosnjaji et al. (2016). An event is described by several heterogeneous fields, numerical and categorical. An essential property of event sequences is that these data are often gathered from multiple sources or channels, rendering *multimodal*.

Thus, the success of financial organizations strongly depends on their availability to analyze such multi-source heterogeneous event sequences accurately. However, existing multimodal models Xu et al. (2023); Zhang & Yan (2023) cannot be directly applied to such event/tabular data due to their significant difference with audio, images, texts, and regular time-series. Unfortunately, despite the urgent business needs, the progress in the development of multimodal techniques for multi-source event sequences is limited by the absence of large-scale datasets Indeed, though several datasets of event sequences are used in research, e.g., credit card transactions Padhi et al. (2021) or MIMIC Johnson et al. (2023), they are either small or contain only one modality. Thus, tackling the complexity of multimodal event sequence data is still very challenging.

To bridge this gap, this paper introduces the Multimodal Banking Dataset (MBD), an unprecedented open-source resource encompassing extensive multichannel event sequence data of banking corporate clients. It is the largest of its kind, featuring detailed records of approximately 2 million clients across four distinct modalities: money transfers (about 950 million events), geo position data (around 1 billion events), technical support dialog embeddings (approximately 5 million entries), and monthly aggregated bank product purchases categorized into four types. Each modality encompasses roughly one or two years of historical, time-annotated data, making it a rich resource for analyzing the dynamics of client behavior over time.

The MBD dataset enables the research of several critical business problems in a multimodal context, such as future purchase prediction (campaigning) and matching of different modalities for the same clients. In addition, we provide benchmarks for these tasks using the MBD and other existing financial datasets of much smaller size. Our possibility to publish the dataset is caused by properly anonymizing all data to protect client privacy. Our experiments confirm that this procedure preserves the consistency of model performance between original and anonymized data.

## 2 RELATED WORKS

### 2.1 FINANCIAL DATA

The multitude of services and processes in banks generates a variety of data that can be considered as modalities. Early works such as Moro et al. (2014); Mancisidor et al. (2021) use feature processing techniques that remove multimodal complexity and present data in tabular form. The Amex dataset ame improves information content and complexity. Here, a wide range of different financial aggregates are presented as a sequence of historical slices. The development of deep learning methods has led to the ability to work with complexly structured data, such as sequences of events. Quite a few datasets age; ros; alp, mostly unimodal, presented mainly at ML competitions. To work with such data, both supervised Ala'raj et al. (2022); Babaev et al. (2019); Wang & Xiao (2022) and unsupervised methods Padhi et al. (2021); Babaev et al. (2022); Skalski et al. (2023) are used. A multimodal financial sequential dataset was introduced in DataFusion 2022 competition dat. There are two sequential modalities, transaction and web clickstream, and two downstream tasks: matching and education level prediction. However, this is an extremely small dataset of 22K clients, and no accurate baseline model is available.

### 2.2 OTHER EVENT SEQUENCE DOMAINS

Temporal point process Mei & Eisner (2017); Zhuzhel et al. (2023) model streams of discrete events in continuous time by constructing a neurally multivariate point process. The authors use a large collection of datasets from different types of modalities: Media (Retweets Zhao et al. (2015), Meme-Track Leskovec & Krevl (2014), Amazon ama (2018), IPTV Luo et al. (2014)), Medical (MIMIC-II Johnson et al. (2016)), Social(Stack Overflow Leskovec & Krevl (2014), Linkedin Xu & Zha (2017)) and Financial (Transaction Fursov et al. (2021)) data. All datasets are independent, and each one is single-modal. EventStreamGPT McDermott et al. (2024) uses multimodal medical record datasets, MIMIC-IV Johnson et al. (2023). The authors propose a GPT-like approach for continuous-time event sequences. The structure of this dataset is close to the MDB. However, financial data, unlike medical data, contains longer chains of events, more regular patterns, and individual transactions are less informative.

### 2.3 GEOSTREAM AND DIALOGUES

Geodata is used for various tasks. One of the uses of geo is a visualization of analytics on a map Hao et al. (2011). However, geo is used here not as a separate modality but as additional tags to the mainstream of tweets. Mobile marketers Baye et al. (2024) use geo-targeting for pricing and send personalized recommendations. In Verma et al. (2020), geo hashes are used for user mobility detection and prediction.

We encode our dialogue entries via a pretrained NLP model. Such models have already been used Hassan et al. (2019) for text anonymization tasks. Embeddings preserve the meaning of the text, which was shown in Vaswani et al. (2017). Pre-trained text embeddings can capture text sentiment and improve text-to-speech models Hayashi et al. (2019).

## 3 PROPOSED DATASET

Modern innovative banking institutions actively develop AI technologies for customizing their human-oriented technologies and making everyday decisions. A superior level of technologies will lead to new cases of customer experience, which should form a competitive advantage of services

provided, including the speed, accuracy, and price of customer services, including personal credit conditions, individual finance strategy, etc.

One of the main benefits of using AI is the ability to analyze large amounts of customer data. This helps banks better understand their customers' needs and offer them the most suitable products and services. Additionally, AI can protect customers from fraud and prevent financial losses. More accurate forecasting of financial risks associated with lending or investing allows us to provide more favorable conditions to more reliable clients. This strategy is more effective if there is a lot of data, so banks strive to accumulate as much information as possible.

Being a team of a bank that stores petabytes of data about bank processes and clients, we understand the urgent needs of fintech data scientists in large sets of publicly available temporal data from various sources (bank transactions, client locations, purchased products, etc.) to drive innovation and scientific discovery. Unfortunately, the number of appropriate datasets is limited because providing such data comes with certain risks. Typically, banks are wary of sharing their data due to potential leaks of confidential information or violations of data protection regulations. In addition, the data is considered to have commercial value, and companies do not want to disclose it. To mitigate these risks, it is required to take all necessary steps to ensure data security and remove any identifying information. This allows information to be shared without violating client confidentiality, but this procedure requires significant effort from engineers, managers, and lawyers. Though there exist a small number of properly anonymized banking datasets, such as credit card transactions Padhi et al. (2021) or AlphaBattle alp, to the best of our knowledge, there are no publicly available large multimodal temporal datasets for banks.

Thus, in this paper, we introduced the first large-scale multimodal banking dataset to support future research on multimodal techniques for event sequences. In particular, we select several practically important tasks, such as campaigning, i.e., prediction if a client would purchase some of four rather popular products in the next month. Each client is described by sequences of typical bank data: transactions, geo positions where the customer used the bank application, and dialogues with technical support. These data sources highlight the main difficulties in developing multimodal models, namely, asynchronous events in different modalities, various intensities of events, rare/irregular events, and even the absence of some modalities for many clients. Based on our dataset, the researchers will be able to fully take into account cross-modal connections of sequences from multiple sources at the level of individual events.

Let us discuss the details of the dataset collection procedure. At first, we select a complete sample of clients for two years (2021 and 2022) to cover all seasons. Among all customers who had the opportunity to purchase at least one of four products during 2022, we randomly choose 2,186,230 clients, among which 1M customers are labeled by monthly aggregated purchases of each of four products in each month. For these clients, we collect 947,899,612 financial operations, 1,117,213,760 geo position events, and 5,080,781 dialogues with technical support. Our data raise typical practical challenges for training multimodal models. For example, many clients do not have all three modalities simultaneously because they can never make a transaction, call tech support, or leave their geo trace while running the bank application. Next, all the data are properly anonymized to guarantee the confidentiality and privacy of customer information. As a result, it is impossible to recover real clients from our anonymous data. We will show in the experimental study that such anonymization still allows us to extract valuable information about clients.

To demonstrate the complete transformation of data from various sources, Fig. 1 briefly shows example data for each modality, and the temporal structure of the data for a campaigning model is presented. Let us introduce further details for each modality in the following subsections.

We present a comparative analysis of various event sequence datasets alongside our MBD dataset. As shown in Table 1, the MBD dataset is substantially more comprehensive, offering a greater number of modalities, events, clients, and downstream tasks compared to other datasets. MBD incorporates a diverse set of data modalities, including bank transactions, geo-locations, and technical support dialogues, providing a richer and more realistic basis for analysis. Additionally, it supports a broader range of downstream tasks, detailed in the following sections, allowing for more sophisticated and flexible modeling approaches.

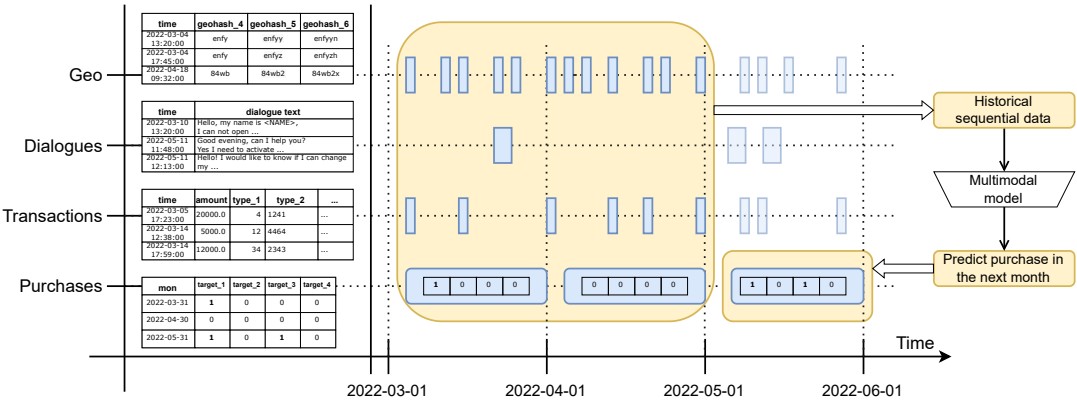

Figure 1: Pipeline for processing original data sources for solving campaigning task. Examples of raw data for each modality are presented on the left. Only anonymized data is published in the dataset. The center shows the temporal anonymized structure of the data. The multimodal multi-label classification model for predicting purchases is shown on the right.

Table 1: Overview of existing transaction datasets.

| Dataset | # Clients | Downstream Tasks | # Events | Class Balance | Modalities |
|---|---|---|---|---|---|
| **Datafusion** dat | 22K | Binary classification Multimodal matching | 146M | Imbalanced | Transactions, Clickstream |
| **Alphabattle** alp | 1.5M | Binary classification | 443M | Imbalanced | Transactions |
| **Age** age | 50K | Multiclass classification | 44M | Balanced | Transactions |
| **Rosbank** ros | 10K | Binary classification | 1M | Imbalanced | Transactions |
| **Credit Card Transaction** Padhi et al. (2021) | 2K | Binary classification Regression task | 2M | Highly Imbalanced | Transactions |
| **MBD (ours)** | 2M | Multilabel binary classification Multimodal matching | 2B | Highly Imbalanced | Transactions, Geostream, Dialogues |

## 3.1 MODALITIES

**1. Bank transactional data** are financial operations (events) carried out between different clients. Collected over a two-year period (2021 and 2022), the sequence of financial operations can uniquely characterize the client Babaev et al. (2022), so this data source plays one of the most significant roles in planning and recommendations. Thus, the main component of our MBD dataset is each client's transactional history, represented by an event with a timestamp and various attributes of the anonymized counterparty. Clients have 638 transactions on average.

**2. Dialogues.** The dialogue data consists of transcriptions from customer calls to technical support and negotiations between clients and their managers, collected over a two-year period (2021 and 2022). We incorporate dialogues from key communication channels, including sales and service calls, which account for most interactions with bank customers. It is an extremely important source of information about client needs and problems Bauman et al. (2024). The audio utterance is fed into a commercial Speech-to-Text algorithm. Personal information, e.g., the client's name, is detected in the text and masked. To further anonymize the dialogue, we feed its text into a pre-trained NLP model[1] and save the resulting embeddings of size 768 in dialogue modality. Only 46% of customers contact support and have records of conversations, 98% of them have no more than 10 dialogues.

**3. Geostream data** contains a sequence of geo-coordinates of a client obtained throughout 2022. To anonymize this modality, the coordinates are encoded using geohashes[2], a geocoding system that converts a geographic location into a short string. Each unique geohash corresponds to a region on the Earth's surface. It is possible to adjust the accuracy and size by removing characters from the end of the code. In our dataset, the coordinates are encoded with a precision of 4, 5, and 6 characters,

---

[1]https://huggingface.co/ai-forever/ruBert-base
[2]https://pypi.org/project/pygeohash/

representing cells of different sizes on the map. As a result, there are 43,999 distinct values of geohash_4, 347,698 numbers of geohash_5, and 2,264,404 most precise locations (geohash_6).

**4. Products purchases**. Our dataset serves as a valuable resource for analyzing the needs of bank customers and optimizing the campaigning process, a critical task that directly impacts both the volume of products sold by the bank and its overall profitability. High-quality recommendations play a key role in enhancing the customer experience, making campaigning essential not only for business outcomes but also for customer satisfaction. Specifically, MBD includes monthly data on the purchases of four distinct banking products throughout 2022, providing a broader temporal scope that captures patterns beyond the pandemic's peak. We concentrate on the most popular of these products, as internal analysis across various tasks using proprietary data consistently showed that these products provide a robust foundation for model selection. The insights from this data enable the development of models that demonstrate superior performance across a broad spectrum of related tasks. To predict a purchase in a certain month, it is necessary to take events (transactions, geo, dialogues) strictly before the beginning of this month. Therefore, the date range for the purchases dataset is shifted by 1 month, i.e., information is available from February 1, 2022, to January 31, 2023. The campaigning task is a multi-label classification problem, i.e., we store a binary label for each product that indicates whether it is purchased by a customer in a certain month. The peculiarity of this dataset is its imbalance, which is specific to this type of business task: 81% of clients have no purchases, 15% have one, and the remaining 4% have two or more purchases. A historical overview over 12 months allows us to model the customer behavior dynamic and predict the date of purchase more accurately.

Detailed information on all modalities is provided in Appendix A, including the sequence length of event sequences and data samples.

### 3.2    DATA ANONYMIZATION

Our dataset contains no personal or confidential information whatsoever. Nevertheless, the event sequences are detailed enough that it could be possible to compare individuals from the publicly accessible portion of the dataset with the original proprietary data. To mitigate this risk, noise is introduced to the data, ensuring that such comparisons and identification are impossible. The noise patterns were selected by our bank's internal security department. These patterns are applied locally, preserving the overall structure of the data. The specific noise parameters are not disclosed to prevent potential attacks on the dataset.

All ID fields are hashed with a random salt. All categorical field values are mapped to enumerated indexes. Random noise is added to numerical fields and dates, preserving the hour of the original date, which may be the cause of the shuffle of the local sequence. The dialogue embedding space is divided into regions, which are then shuffled.

## 4    BENCHMARK

In this paper, we introduce a benchmark for widely used event sequence datasets, incorporating practically important downstream tasks. This section provides a detailed description of the downstream tasks, baseline methods, and evaluation protocols.

### 4.1    DATASETS AND DOWNSTREAM TASKS

For each downstream task in every dataset, we implement an out-of-fold validation protocol to conduct our experiments. The client dataset is partitioned into five folds, with four folds used for training and the remaining fold reserved for testing. The training and testing sets are publicly available alongside the dataset, allowing future researchers to compare performance metrics. As each dataset in our benchmark exhibits label imbalance, ROC AUC is the most robust and informative evaluation metric due to its resilience to class imbalances. In real-world business, campaign effectiveness is measured by revenue, but conducting A/B tests for every ML model is impractical. Instead, ROC AUC is a reliable proxy metric for model comparison, with only the top performers advancing to A/B testing. This method was validated through real-world A/B tests and is recognized as the core evaluation metric by leading institutions, including one of the largest global banks. The choice

of AUC for campaigning is supported by the reason that ranking models by their ROC curves are similar to comparing their non-response ratio at all possible cutoff points simultaneously Liu et al. (2012); Rosset et al. (2001). Let us discuss the details of downstream tasks for each dataset in our benchmark.

**1. MBD.** For our dataset, we introduce a campaigning downstream task. In this task, it is required to predict the customer's propensity to purchase four different products in the next month (Fig. 1) given sequences of transactions, geo locations, and dialogues from the beginning of this month. Solutions to this problem are used to plan marketing campaigns and prepare sales communications through various communication channels with the client.

The baseline methods outlined in the following section are applied as follows. First, we train our models using the training set. Considering the temporal structure of our target, we compute the embedding of each client's history for up to one month, focusing on the presence of the target product (Fig. 1). We then evaluate the model using the multi-label classification metric ROC AUC across the 12 months of 2022 and for four binary product labels.

**2. Datafusion.** In this dataset, the proposed downstream task is to predict the higher education attainment of bank clients. It involves analyzing two client modalities (transaction histories and clickstream data) to accurately infer their educational background. The task is formulated as a binary classification problem, with 75% of the labels corresponding to clients with higher education. The objective is to develop predictive models that leverage these multimodal data sources to extract meaningful insights, which can be applied to further analysis.

**3. Alphabattle.** We incorporate the large unimodal Alphabattle dataset into our benchmark alongside the multimodal datasets. This inclusion of data from various financial institutions aims to support more robust and reliable conclusions. In this dataset, the downstream task estimates the probability of a customer defaulting based on their historical card transaction behavior. This task is framed as a binary classification problem, with 2.7% of the labels representing clients who have defaulted. Although the dataset is unimodal, the downstream task remains highly relevant for financial institutions, offering critical insights into credit risk assessment in improving decision-making processes related to customer management and financial strategies.

**Multimodal matching**

For the multimodal datasets MBD and Datafusion, we propose a downstream task of multimodal matching Zong et al. (2023). Multimodal matching involves aligning and comparing modalities to identify meaningful relationships or connections. Frequently, data from multiple sources for the same client are matched using predefined rules or heuristics, which may not always yield optimal results. To enhance the accuracy of this process, specialized identification algorithms are required to compare modalities more precisely.

For the matching task, we employ a framework analogous to CLIP Radford et al. (2021). We utilize GRU encoders to embed pairs of samples from two input modalities, labeling them as either positive (i.e., data from the same client) or negative matches (i.e., data from different clients). The model is trained using the InfoNCE loss function Chen et al. (2020), which maximizes similarity for positive pairs while minimizing it for negative pairs. To assess the model's performance, we use Recall@1, Recall@50 and Recall@100 metrics.

### 4.2 METHODS

To establish performance baselines, we implement several widely adopted architectures. Our approach prioritizes unsupervised and semi-supervised methods Balestriero et al. (2023), enabling the training of a general-purpose encoder on unlabeled sequential data. Additionally, we incorporate supervised methods that allow for the immediate training of the encoder in a fully supervised manner.

#### 4.2.1 UNIMODAL APPROACHES

The following techniques are implemented in our benchmark to extract features from data:

1. **Aggregation Baseline** that contains hand-crafted aggregation statistics Babaev et al. (2022): Events are represented either numerically, such as transaction amount, or cate-

gorically, like event types. For numerical attributes, we apply aggregation functions (e.g., sum, mean, std) across all events in a sequence. Categorical attributes are grouped by unique values, aggregating numerical attributes using functions like count or mean.

2. **CoLES** (Contrastive Learning for Event Sequences), a self-supervised contrastive model Babaev et al. (2022) specially developed to obtain representations of such event sequences as bank transactions. The sequence encoder is a GRU (Gated Recurrent Unit) with a hidden size of 256.

3. Two **Tabular Transformers** from IBM Padhi et al. (2021). The first model, TabBERT, adapts BERT to event sequences such as bank transactions. The second model, TabGPT, was initially proposed to generate synthetic tabular sequences. Both models extract 256-dimensional embeddings of an input event sequence. After that, we pool output embeddings of the client in result embedding of size 1024, calculating min, max, mean, and std.

To obtain representation of a sequence of **dialogues**, we borrow several conventional techniques to aggregate the sequence of embeddings of each dialog of a client: 1) mean pooling of all embeddings; and 2) use only the most recent embedding for the date of interest.

For unimodal supervised methods (Supervised RNN), we utilize GRU architectures Babaev et al. (2019) with a hidden size 32. The models are trained in a multi-label setting using binary cross-entropy (BCE) loss, ensuring effective optimization for tasks with multiple targets.

### 4.2.2 MULTIMODAL APPROACHES

To explore the potential of **multimodal processing** for event sequence analysis, we compare several fusion techniques:

1. **Blending** computes a weighted sum of class posterior probabilities from individual single-modal classifiers, effectively combining the predictions from each modality.

2. **Late Fusion**: embeddings from all data sources are concatenated and fed into a classifier. This technique allows the model to learn interactions between modalities after they have been individually processed Huang et al. (2020). In supervised Late Fusion, we utilize separate GRU encoders for each modality, concatenating their embeddings to form a unified representation (Supervised RNN).

3. **Early Fusion** combines representations from multiple modalities at the initial stages of the model, enabling joint processing of multimodal data. We employ the CrossTransformer approach Zhang & Yan (2023), which utilizes a cross-attention mechanism to integrate information across modalities efficiently. In our experiments, this method is applied within a supervised learning framework.

## 5 EXPERIMENTS

Our models, experiments, and training procedures were implemented in Python, leveraging PyTorch and PyTorch Lightning for deep learning tasks, and PySpark for distributed data processing. We trained the neural networks using NVIDIA V100 GPU, while the boosting models were trained on computational clusters equipped with 600 cores. The reported experiments, including extensive hyperparameters optimization, required approximately 500 hours of computation.

### 5.1 MODEL AND TRAINING HYPERPARAMETERS

We employ the unsupervised baseline methods (CoLES, TabGPT, TabBERT, Aggregation) with default hyperparameters from the pytorch-lifestream framework. The PyTorch implementation of the Adam optimizer is utilized, with an initial learning rate of 0.001, coupled with the StepLR scheduler. Models are saved based on the lowest validation loss or the highest validation unsupervised metrics Tsitsulin et al. (2023), evaluated after each training epoch, with training of 15 epochs. We employ 24-dimensional embeddings for categorical features and clipped the number of categories for features with many unique values. We apply either an identical mapping or a logarithmic transformation for numerical features. We use the gradient boosting algorithm available in PySpark ML for downstream tasks.

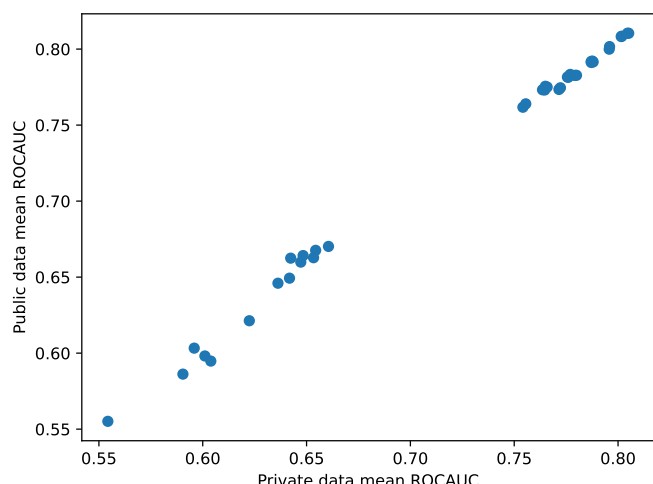

Figure 2: Model performance comparison on private and public data. Kendall-Tau=0.94

## 5.2 DOWNSTREAM TASKS

### 5.2.1 MBD: CAMPAIGNING TASK

This Subsection contains experimental results for the campaigning task in the MBD dataset in unimodal and multimodal baselines. One of the main objectives of our paper is to provide a real data benchmark to facilitate the development of multimodal algorithms. To achieve this, it is necessary to demonstrate that an algorithm outperforming another on our public benchmark will similarly outperform it on real data.

Table 2 shows that the transaction modality plays the most crucial role in achieving accurate classification. In contrast, dialogues and geostream, when used in an unimodal setting, perform only slightly better than a random estimator. However, as shown in Table 3, the predictive performance improved significantly in the multimodal setting by integrating additional modalities. The overall trend indicates a consistent improvement in validation metrics as more modalities are incorporated. Specifically, the multimodal late fusion approach enhances predictive accuracy by 1-1,5% when adding other data sources to the transaction stream.

Fig. 2 highlights a strong correlation between performance metrics on public and private datasets, with a Kendall-tau correlation coefficient of 0.94. Hence, the anonymization process has minimal impact on model performance for the downstream task of campaigning. The consistency in relative ranking across both datasets underscores the reliability of our benchmark for advancing research in multimodal event sequence analysis.

More detailed comparison of result on MBD and the private dataset and comprehensive results for each modality and all possible fusion combinations of multiple modalities are shown in Appendix B (Tables 9 - 12). Our modalities, namely, geostream, transactions, and dialogues, are denoted as Geo, Trx, and Dialogs, respectively. To specify a method applied to a modality, we use a clear notation. For instance, if embeddings from the CoLES model are applied to transactions, we denote this as TrxCoLES.

### 5.2.2 DATAFUSION: HIGHER EDUCATION

In this subsection, we present the results of unimodal and multimodal experiments on the DataFusion dataset, as shown in Tables 2 and 3. While the dataset is small, leading to an insignificant increase in quality metrics when incorporating additional sources in multimodal settings, the results in both unimodal and multimodal configurations remain valuable as they contribute to expanding our benchmark.

Table 2: Mean ROC-AUC of downstream results using unimodal methods.

| Model | MBD | | Datafusion | | Alphabattle |
|---|---|---|---|---|---|
| | Transactions | Geostream | Transactions | Clickstream | Transactions |
| Aggregation | $0.783 \pm 0.002$ | $0.595 \pm 0.002$ | $0.793 \pm 0.013$ | $0.537 \pm 0.018$ | $0.785 \pm 0.0010$ |
| CoLES | $0.773 \pm 0.002$ | $0.598 \pm 0.004$ | $0.784 \pm 0.012$ | $0.641 \pm 0.013$ | $0.793 \pm 0.0005$ |
| TabBERT | $0.762 \pm 0.004$ | $0.603 \pm 0.002$ | $0.762 \pm 0.014$ | $0.590 \pm 0.026$ | $0.778 \pm 0.0003$ |
| TabGPT | $0.802 \pm 0.002$ | $0.621 \pm 0.003$ | $0.766 \pm 0.013$ | $0.618 \pm 0.016$ | $0.775 \pm 0.0010$ |
| Supervised RNN | $0.819 \pm 0.002$ | $0.540 \pm 0.012$ | $0.712 \pm 0.016$ | $0.563 \pm 0.011$ | $0.792 \pm 0.0030$ |

Table 3: Mean ROC-AUC in late fusion setting.

| Dataset | Modalities | CoLES | TabGPT | TabBERT | Supervised RNN |
|---|---|---|---|---|---|
| MBD | Trx | $0.773 \pm 0.002$ | $0.802 \pm 0.001$ | $0.762 \pm 0.004$ | $0.819 \pm 0.002$ |
| | Trx + Geo | $0.775 \pm 0.002$ | $0.800 \pm 0.001$ | $0.764 \pm 0.004$ | $0.819 \pm 0.001$ |
| | Trx + Dialog | $0.781 \pm 0.002$ | $0.810 \pm 0.002$ | $0.773 \pm 0.003$ | $0.821 \pm 0.0006$ |
| | Trx + Dialog + Geo | $0.783 \pm 0.002$ | $0.808 \pm 0.001$ | $0.775 \pm 0.003$ | $0.824 \pm 0.001$ |
| Datafusion | Trx | $0.784 \pm 0.012$ | $0.766 \pm 0.013$ | $0.762 \pm 0.014$ | $0.712 \pm 0.015$ |
| | Trx + Click | $0.785 \pm 0.011$ | $0.766 \pm 0.011$ | $0.761 \pm 0.012$ | $0.703 \pm 0.008$ |

### 5.2.3 ALPHABATTLE: DEFAULT

In Table 2, we present the metrics for unimodal methods on the Alphabattle dataset. Interestingly, the results demonstrate that supervised methods on both the MBD and Alphabattle datasets perform as well as, or better than, unsupervised methods. This may be attributed to the dataset size and the amount of labeled data available.

### 5.2.4 MULTIMODAL MATCHING TASK

In this subsection, we present the results of our proposed multimodal matching benchmark, summarized in Table 4, which includes both MBD and Datafusion datasets. Detailed results for other modalities can be found in Appendix B, in Table 7. We report Recall@1, Recall@50, and Recall@100 for each modality, measured in both directions. For example, in the case of the transaction and geostream pair for MBD, we compute both Trx2Geo and Geo2Trx, allowing for an evaluation of alignment from both perspectives.

For MBD, our analysis reveals considerable variation in performance across different modality pairs. Specifically, dialogue data consistently exhibits weaker matching performance compared to other modalities, such as transactions and geostream, demonstrating significantly stronger alignment. This disparity suggests potential limitations within the dialogue modality, indicating that it may offer less complementary or aligned information. Alternatively, the unique structure of dialogue data may necessitate more sophisticated or specialized techniques for effective integration with other modalities. We also present the multimodal matching results for the DataFusion dataset.

### 5.3 COMPARISON OF MULTIMODAL FUSION METHODS

In this subsection, we evaluate the performance of multimodal fusion techniques across multiple datasets. For this analysis, we select the best-performing models for Blending and Late Fusion to compare against Early Fusion. As shown in Table 5, Late Fusion with a Supervised RNN consistently delivers superior results on the MBD dataset, demonstrating its robustness in effectively integrating multimodal data. Early Fusion, implemented via the cross-attention from CrossTransformer Zhang & Yan (2023), achieves competitive performance but slightly lags behind Late Fusion while Blending exhibits considerably weaker results. On the Datafusion dataset, Late Fusion with TabGPT is the most effective method, with Early Fusion also performing well and outperforming Late Fusion with a Supervised RNN. Blending, in contrast, consistently yields the lowest performance across all modalities. These findings highlight the effectiveness and reliability of Late Fusion for multimodal data integration while identifying Early Fusion as a promising direction for further research and optimization.

Table 4: Multimodal matching task

| Dataset | Modalities | Recall@1 | Recall@50 | Recall@100 |
|---------|-----------|----------|-----------|------------|
| **MBD** | Trx2Geo | $0.006 \pm 0.0003$ | $0.196 \pm 0.002$ | $0.303 \pm 0.004$ |
|  | Geo2Trx | $0.004 \pm 0.0003$ | $0.162 \pm 0.002$ | $0.262 \pm 0.004$ |
| **Datafusion** | Trx2Click | $0.002 \pm 0.0010$ | $0.063 \pm 0.004$ | $0.120 \pm 0.009$ |
|  | Click2Trx | $0.001 \pm 0.0007$ | $0.070 \pm 0.005$ | $0.115 \pm 0.008$ |

Table 5: Comparison of Multimodal Fusion Techniques: Blending, Late Fusion, and Early Fusion.

| Dataset | Modalities | Blending TabGPT | Late Fusion | | Early Fusion CrossTransformer |
|---------|-----------|-----------------|-------------|---|-------------------------------|
|  |  |  | **TabGPT** | **Supervised RNN** |  |
| **MBD** | Trx + Geo | $0.804 \pm 0.001$ | $0.800 \pm 0.001$ | $0.819 \pm 0.001$ | $0.815 \pm 0.001$ |
|  | Trx + Dialog | $0.742 \pm 0.001$ | $0.810 \pm 0.002$ | $0.821 \pm 0.0006$ | $0.821 \pm 0.002$ |
| **Datafusion** | Trx + Click | $0.756 \pm 0.013$ | $0.766 \pm 0.011$ | $0.703 \pm 0.008$ | $0.735 \pm 0.010$ |

## 6 LIMITATIONS

Data was subject to de-identification, which limits the possibility of using models trained on this dataset outside of it. Also, the data analysis results can not be generalized. In other words, based on this dataset, it is impossible to draw conclusions regarding specific regions and market characteristics or perform deep text analytics. However, within our benchmark, the data is consistent, which allows us to draw correct conclusions about the performance of multimodal or unimodal methods for working with sequences. It is also worth noting that the study was conducted on a sample of clients of a certain segment who had the opportunity to purchase certain products and does not cover all possible groups of consumers.

## 7 CONCLUSION AND FUTURE WORK

In this paper, we present the first large-scale, publicly available multimodal banking dataset, MBD, which comprises anonymized sequential data, including bank transactions, geo-locations, and technical support dialogues for more than 2 million bank clients. Our findings indicate that anonymization does not significantly affect algorithm performance, making the dataset ideal for selecting models suitable for deployment in real-world production environments. Furthermore, excluding sensitive attributes such as gender, age, and race mitigates the potential for bias in the resulting models, promoting the development of more ethical AI systems.

Moreover, MBD, together with the Datafusion and Alphabattle datasets, serves as the foundation for a novel benchmark targeting key practical downstream tasks. This benchmark paves the way for the development of scalable algorithms, both multimodal and unimodal, with potential applications in event sequence prediction across various industries. Our experimental results demonstrate that even basic multimodal fusion techniques surpass single-modal baselines in overall model quality (Table 3). Furthermore, based on our results of Late and Early Fusion (Table 5), it is possible to develop more advanced models that can effectively capture interactions between modalities, to achieve further improvements in overall performance. Given the scale of data and its real-world applicability, even moderate improvements in model performance metrics can translate into substantial financial benefits when applied to a large customer base.

In the future, we are going to extend the dataset to its new versions. First, it is necessary to incorporate new data sources, which will predominantly be utilized as additional data sources (e.g., clickstreams or enriched dialogue features, e.g., discussed topics). Second, it is important to expand a set of downstream tasks to campaigning for other financial products and analyzing customer behavior insights (customer churn, fraud detection, credit risk assessment). Finally, it is possible to extend the time range of the dataset, enabling longitudinal studies and trend analysis.

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

## A  DATASET STATISTICS AND DATA SAMPLES

In this section, we present data samples from various modalities and a histogram of sequence lengths. Table 6 provides a list of attributes for the transactional modalities, while Figure 3 illustrates the distribution of sequence lengths for these modalities. Additionally, Figure 6 shows a sample of geographical data, and Figure 7 displays the distribution of sequence lengths for geostream modalities. For dialogue data, a sample is presented in Figure 4, with the distribution of sequence lengths for dialogue events shown in Figure 5.Figure 8 demonstrates a sample of data for product purchases. Here, we observe that geographical and transactional modalities have long tails in the histogram of distributions.

Table 6: Transaction features

| Features | Number of categories | Description |
|---|---|---|
| amount | - | transaction amount |
| event time | - | transaction time |
| currency | 15 | transfer currency |
| event type | 56 | transaction type |
| event subtype | 62 | transaction subtype |
| src type11 | 101 | sender field type 1 |
| src type12 | 536 | sender field subtype 1 |
| dst type11 | 123 | receiver field type 1 |
| dst type12 | 649 | receiver field subtype 1 |
| src type21 | 40131 | sender field type 2 |
| src type22 | 87 | sender field subtype 2 |
| src type31 | 2389 | sender field type 3 |
| src type32 | 88 | sender field subtype 3 |

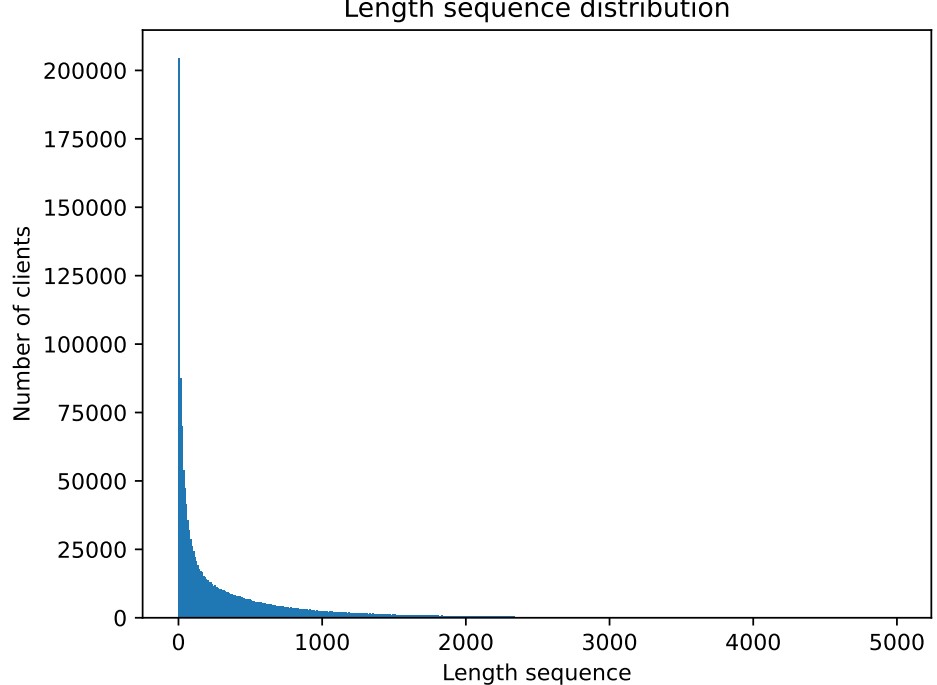

Figure 3: The histogram of the number of clients with a certain length of transaction history

## B DETAILED EXPERIMENTAL RESULTS

### B.1 MULTIMODAL MATCHING

For the multimodal matching task Table 7, it is observed that the dialogue modality exhibits poor compatibility when combined with other modalities.

### B.2 HANDLING CLASS IMBALANCE

We conducted experiments on the MBD dataset, focusing on transaction modalities, and explored various techniques including random undersampling, oversampling, and class balancing within gradient boosting for CoLES embeddings. Additionally, stratified batching was applied to better align

| client_id | event_time | embedding |
|---|---|---|
| a039ad3b595d4f5b1a… | 2022-05-02 12:24:30…. | [-3.24845104e-03  1.40231192e-01  5.8907… |
| a039ad3b595d4f5b1a… | 2022-10-28 09:05:18…. | [ 5.89274131e-02 -7.72307068e-03  2.3162… |
| a039ad3b595d4f5b1a… | 2022-12-15 08:36:26…. | [ 7.98072889e-02 -3.91234783e-03  2.8684… |
| a060e69e9e049ad01… | 2022-09-13 13:37:03…. | [ 0.5237517  -0.3054205   0.6801795  -0.57… |
| a08c690dd972d2066… | 2022-05-02 09:12:22…. | [-9.23508126e-03 -6.97142631e-02  2.1549… |

Figure 4: Sample data of dialogues modality.

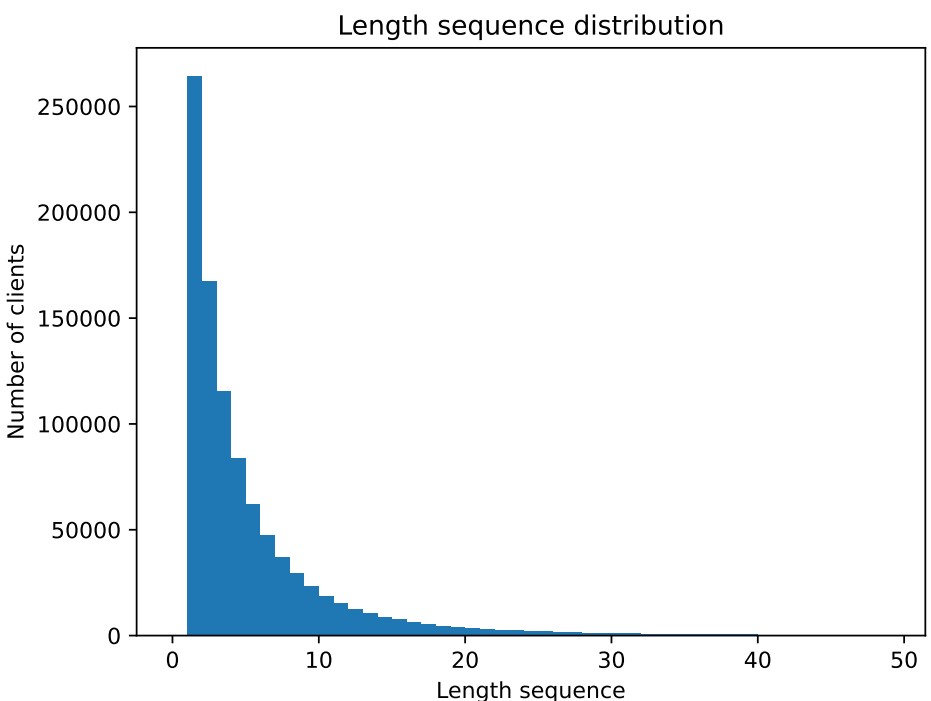

Figure 5: The histogram of the number of clients with a certain number of dialogues

| client_id | event_time | geohash_4 | geohash_5 | geohash_6 |
|---|---|---|---|---|
| 309c0e909835757db… | 2022-08-27 09:56:36…. | 39879 | 144891 | 1959174 |
| 309c0e909835757db… | 2022-08-14 07:13:23…. | 39879 | 144891 | 1959174 |
| 309c0e909835757db… | 2022-08-02 07:46:18…. | 39879 | 144891 | 1959174 |
| 309c0e909835757db… | 2022-08-19 08:47:39…. | 39879 | 144891 | 1959174 |
| 309c0e909835757db… | 2022-08-19 10:15:14…. | 39879 | 144891 | 1959174 |

Figure 6: Sample data of geostream modality.

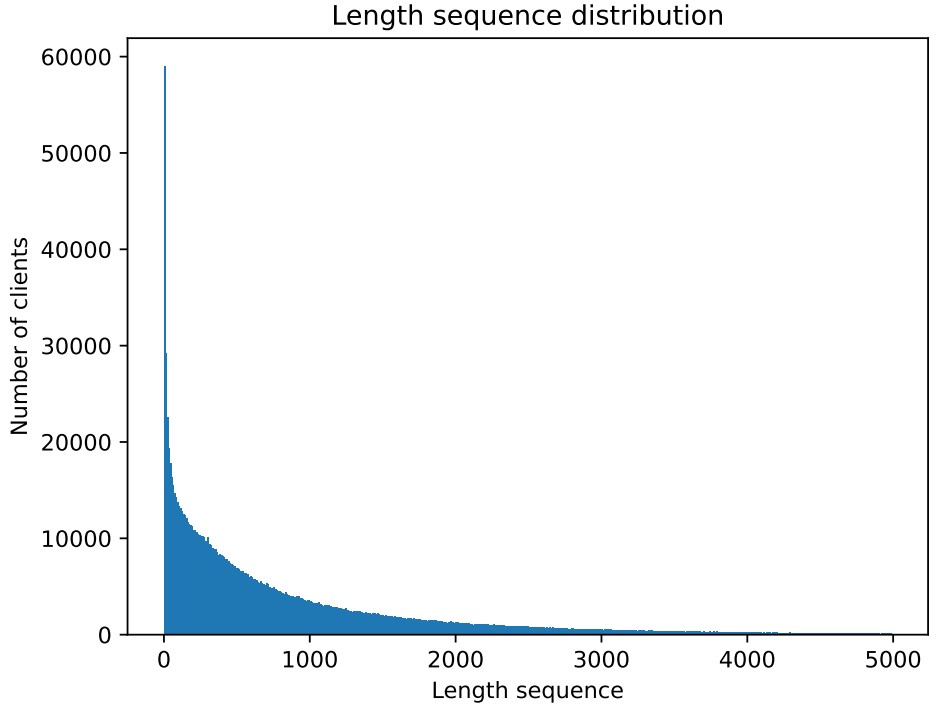

Figure 7: The histogram of the number of clients with a certain length of geostream

| mon | target_1 | target_2 | target_3 | target_4 | client_id |
|---|---|---|---|---|---|
| 2022-06-30 | 0 | 0 | 0 | 0 | 1d4ebf30ab5b981c60... |
| 2022-07-31 | 1 | 0 | 0 | 0 | 1d4ebf30ab5b981c60... |
| 2022-08-31 | 0 | 0 | 0 | 0 | 1d4ebf30ab5b981c60... |
| 2022-09-30 | 0 | 0 | 0 | 0 | 1d4ebf30ab5b981c60... |
| 2022-10-31 | 0 | 1 | 0 | 1 | 1d4ebf30ab5b981c60... |

Figure 8: Example data of client's purchases.

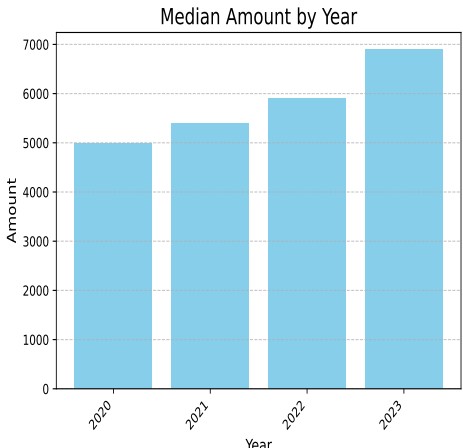
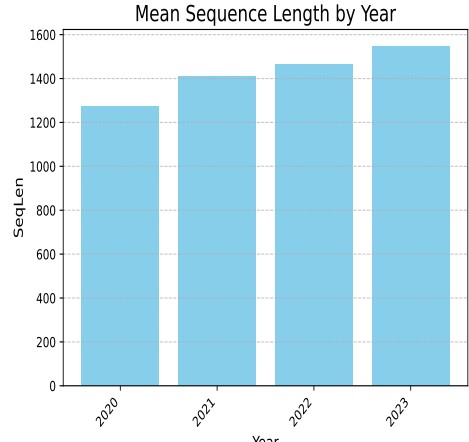

Figure 9: **Median amount per client for private transaction data.** The data reveals a trend of increasing median amounts over four years.

Figure 10: **Mean sequence length per clients for private transaction data.** The mean sequence length has slightly increased over the past four years

Table 7: MBD: Detailed multimodal matching results

|          | Recall@1              | Recall@50            | Recall@100         |
|----------|-----------------------|----------------------|--------------------|
| Trx2Geo  | $0.006 \pm 0.0003$    | $0.1967 \pm 0.002$   | $0.303 \pm 0.004$  |
| Geo2Trx  | $0.004 \pm 0.0003$    | $0.1624 \pm 0.002$   | $0.262 \pm 0.004$  |
| Trx2Dial | $0.00004 \pm 0.00001$ | $0.0001 \pm 0.00005$ | $0.003 \pm 0.0001$ |
| Dial2Trx | $0.00003 \pm 0.00001$ | $0.001 \pm 0.00005$  | $0.003 \pm 0.0001$ |
| Dial2Geo | $0.00002 \pm 0.000001$| $0.0006 \pm 0.00001$ | $0.001 \pm 0.0001$ |
| Geo2Dial | $0.00002 \pm 0.000001$| $0.0005 \pm 0.00001$ | $0.001 \pm 0.0001$ |

the target distribution in Supervised RNN. The results of these experiments are summarized in Table 8.

Here, stratified batching consistently maintains a high ROC-AUC of 0.819 for Supervised RNN, demonstrating its robustness to label imbalance. For CoLES embeddings, balancing techniques result in minimal variations in performance, with ROC-AUC values ranging from 0.772 to 0.774. This indicates that CoLES embeddings exhibit limited sensitivity to label imbalance, highlighting the potential need for more advanced balancing strategies to achieve further improvements.

### B.3 CAMPAINING RESULTS ON THE PRIVATE DATASET AND MBD

The experimental results presented in Tables 11 and 12 underscore the pivotal importance of modality fusion in enhancing model performance. Here, integrating dialogue data with transaction data leads to a significant performance boost. For instance, augmenting TrxTabGPT with DialogLast results in a 1.8% improvement in the mean metric for blending and a 3.4% enhancement in fusion. The most substantial performance gains are observed when all modalities are combined. Specifically, the integration of TrxTabGPT, GeoTabGPT, and DialogLast yields a 2% improvement over unimodal transaction models and a 3.2% improvement in late fusion, highlighting the synergistic benefits of incorporating dialogue and geographical data into transaction-based models. These findings provide robust evidence supporting the effectiveness of multimodal integration. The inclusion of dialogue and geographical data significantly boosts the performance of models centered on transaction data. Moreover, this trend observed in public datasets is consistently replicated in proprietary data, as shown in Tables 9 and 10.

These results also prove the low impact of our anonymization procedure. Indeed, the ranking of methods remains consistent despite differences in absolute metric values. As shown in Table 10,

Table 8: Performance Comparison for Label Imbalance on MBD (Transactions)

| Method | Description | ROC-AUC |
|---|---|---|
| **Supervised RNN** | Baseline | 0.819 |
| **Supervised RNN (stratified batching)** | Balanced target distribution | 0.819 |
| **CoLES** | Baseline embeddings | 0.773 |
| **CoLES (undersampling)** | Random undersampling applied | 0.772 |
| **CoLES (oversampling)** | Random oversampling applied | 0.774 |
| **CoLES (class balanced)** | Class balancing in gradient boosting | 0.774 |

Table 9: Blending results on private dataset

| methods | mean | target_1 | target_2 | target_3 | target_4 |
|---|---|---|---|---|---|
| DialogLast | $0.590 \pm 0.001$ | $0.633 \pm 0.001$ | $0.604 \pm 0.005$ | $0.549 \pm 0.002$ | $0.576 \pm 0.002$ |
| DialogLast+GeoAggregation | $0.603 \pm 0.002$ | $0.632 \pm 0.001$ | $0.629 \pm 0.007$ | $0.558 \pm 0.001$ | $0.592 \pm 0.002$ |
| DialogLast+GeoCoLES | $0.632 \pm 0.002$ | $0.641 \pm 0.001$ | $0.688 \pm 0.008$ | $0.580 \pm 0.002$ | $0.619 \pm 0.003$ |
| DialogLast+GeoTabGPT | $0.648 \pm 0.002$ | $0.654 \pm 0.001$ | $0.715 \pm 0.008$ | $0.591 \pm 0.003$ | $0.632 \pm 0.003$ |
| DialogLast+GeoTabBERT | $0.628 \pm 0.001$ | $0.641 \pm 0.002$ | $0.683 \pm 0.006$ | $0.579 \pm 0.002$ | $0.610 \pm 0.006$ |
| DialogLast+TrxAggregation | $0.731 \pm 0.001$ | $0.701 \pm 0.001$ | $0.816 \pm 0.003$ | $0.673 \pm 0.001$ | $0.735 \pm 0.002$ |
| DialogLast+TrxAggregation+GeoAggregation | $0.729 \pm 0.001$ | $0.694 \pm 0.001$ | $0.812 \pm 0.003$ | $0.670 \pm 0.001$ | $0.738 \pm 0.003$ |
| DialogLast+TrxCoLES | $0.715 \pm 0.002$ | $0.692 \pm 0.002$ | $0.800 \pm 0.005$ | $0.645 \pm 0.002$ | $0.723 \pm 0.005$ |
| DialogLast+TrxCoLES+GeoCoLES | $0.720 \pm 0.002$ | $0.689 \pm 0.003$ | $0.815 \pm 0.003$ | $0.644 \pm 0.002$ | $0.734 \pm 0.005$ |
| DialogLast+TrxTabGPT | $0.739 \pm 0.001$ | $0.683 \pm 0.001$ | $0.820 \pm 0.005$ | $0.685 \pm 0.001$ | $0.767 \pm 0.005$ |
| DialogLast+TrxTabGPT+GeoTabGPT | $0.749 \pm 0.002$ | $0.690 \pm 0.001$ | $0.838 \pm 0.005$ | $0.687 \pm 0.001$ | $0.780 \pm 0.005$ |
| DialogLast+TrxTabBERT | $0.710 \pm 0.006$ | $0.686 \pm 0.004$ | $0.806 \pm 0.004$ | $0.628 \pm 0.006$ | $0.720 \pm 0.012$ |
| DialogLast+TrxTabBERT+GeoTabBERT | $0.716 \pm 0.005$ | $0.683 \pm 0.003$ | $0.821 \pm 0.003$ | $0.627 \pm 0.006$ | $0.731 \pm 0.009$ |
| DialogMean | $0.604 \pm 0.001$ | $0.636 \pm 0.002$ | $0.629 \pm 0.001$ | $0.564 \pm 0.001$ | $0.587 \pm 0.001$ |
| DialogMean+GeoAggregation | $0.613 \pm 0.001$ | $0.635 \pm 0.002$ | $0.648 \pm 0.001$ | $0.568 \pm 0.001$ | $0.601 \pm 0.002$ |
| DialogMean+GeoCoLES | $0.638 \pm 0.002$ | $0.643 \pm 0.002$ | $0.699 \pm 0.005$ | $0.585 \pm 0.001$ | $0.626 \pm 0.001$ |
| DialogMean+GeoTabGPT | $0.653 \pm 0.001$ | $0.656 \pm 0.001$ | $0.725 \pm 0.005$ | $0.595 \pm 0.002$ | $0.638 \pm 0.003$ |
| DialogMean+GeoTabBERT | $0.635 \pm 0.002$ | $0.643 \pm 0.002$ | $0.696 \pm 0.004$ | $0.584 \pm 0.001$ | $0.617 \pm 0.006$ |
| DialogMean+TrxAggregation | $0.732 \pm 0.001$ | $0.700 \pm 0.001$ | $0.816 \pm 0.003$ | $0.673 \pm 0.001$ | $0.739 \pm 0.001$ |
| DialogMean+TrxAggregation+GeoAggregation | $0.729 \pm 0.001$ | $0.694 \pm 0.001$ | $0.812 \pm 0.004$ | $0.670 \pm 0.001$ | $0.741 \pm 0.002$ |
| DialogMean+TrxCoLES | $0.715 \pm 0.002$ | $0.692 \pm 0.002$ | $0.800 \pm 0.004$ | $0.645 \pm 0.002$ | $0.725 \pm 0.004$ |
| DialogMean+TrxCoLES+GeoCoLES | $0.721 \pm 0.002$ | $0.689 \pm 0.003$ | $0.813 \pm 0.003$ | $0.644 \pm 0.002$ | $0.737 \pm 0.004$ |
| DialogMean+TrxTabGPT | $0.739 \pm 0.001$ | $0.682 \pm 0.001$ | $0.819 \pm 0.005$ | $0.684 \pm 0.001$ | $0.769 \pm 0.004$ |
| DialogMean+TrxTabGPT+GeoTabGPT | $0.748 \pm 0.001$ | $0.689 \pm 0.001$ | $0.837 \pm 0.004$ | $0.686 \pm 0.001$ | $0.782 \pm 0.004$ |
| DialogMean+TrxTabBERT | $0.711 \pm 0.006$ | $0.685 \pm 0.004$ | $0.806 \pm 0.005$ | $0.629 \pm 0.005$ | $0.726 \pm 0.011$ |
| DialogMean+TrxTabBERT+GeoTabBERT | $0.717 \pm 0.005$ | $0.682 \pm 0.003$ | $0.821 \pm 0.003$ | $0.628 \pm 0.006$ | $0.735 \pm 0.009$ |
| GeoAggregation | $0.554 \pm 0.001$ | $0.540 \pm 0.001$ | $0.584 \pm 0.002$ | $0.534 \pm 0.001$ | $0.559 \pm 0.001$ |
| GeoCoLES | $0.601 \pm 0.004$ | $0.565 \pm 0.004$ | $0.668 \pm 0.011$ | $0.571 \pm 0.003$ | $0.600 \pm 0.003$ |
| GeoTabGPT | $0.622 \pm 0.001$ | $0.589 \pm 0.001$ | $0.700 \pm 0.008$ | $0.586 \pm 0.003$ | $0.615 \pm 0.004$ |
| GeoTabBERT | $0.596 \pm 0.002$ | $0.566 \pm 0.003$ | $0.663 \pm 0.010$ | $0.570 \pm 0.003$ | $0.585 \pm 0.007$ |
| TrxAggregation | $0.783 \pm 0.001$ | $0.743 \pm 0.001$ | $0.825 \pm 0.002$ | $0.764 \pm 0.001$ | $0.801 \pm 0.002$ |
| TrxAggregation+GeoAggregation | $0.774 \pm 0.001$ | $0.733 \pm 0.001$ | $0.817 \pm 0.003$ | $0.756 \pm 0.001$ | $0.789 \pm 0.003$ |
| TrxCoLES | $0.772 \pm 0.002$ | $0.734 \pm 0.003$ | $0.813 \pm 0.004$ | $0.747 \pm 0.002$ | $0.793 \pm 0.003$ |
| TrxCoLES+GeoCoLES | $0.772 \pm 0.002$ | $0.729 \pm 0.003$ | $0.825 \pm 0.006$ | $0.740 \pm 0.002$ | $0.795 \pm 0.003$ |
| TrxTabGPT | $0.796 \pm 0.000$ | $0.746 \pm 0.001$ | $0.837 \pm 0.004$ | $0.778 \pm 0.001$ | $0.825 \pm 0.004$ |
| TrxTabGPT+GeoTabGPT | $0.798 \pm 0.001$ | $0.743 \pm 0.001$ | $0.850 \pm 0.003$ | $0.772 \pm 0.001$ | $0.827 \pm 0.004$ |
| TrxTabBERT | $0.754 \pm 0.011$ | $0.707 \pm 0.019$ | $0.815 \pm 0.006$ | $0.717 \pm 0.012$ | $0.778 \pm 0.012$ |
| TrxTabBERT+GeoTabBERT | $0.758 \pm 0.010$ | $0.707 \pm 0.016$ | $0.831 \pm 0.006$ | $0.713 \pm 0.011$ | $0.781 \pm 0.010$ |

TrxTabGPT (ROC-AUC 0.796) and TrxAggregation (ROC-AUC 0.780) achieve the best performance on the private dataset. Similarly, TrxTabGPT leads on the public dataset with a ROC-AUC of 0.802 (Table 12). Incorporating geolocation and dialogue modalities further improves results, with DialogLast+TrxTabGPT+GeoTabGPT attaining the highest ROC-AUC on both datasets: 0.802 on the private dataset (Table 10) and 0.808 on the public dataset (Table 12). Overall, multimodal approaches utilizing TabGPT or Aggregation demonstrate superior performance, with Late Fusion consistently outperforming Blending across private and public datasets (compare results in Table 9 and Table 11 to those in Table 10 and Table 12).

Table 10: Late Fusion results on private dataset

| methods | mean | target_1 | target_2 | target_3 | target_4 |
|---|---|---|---|---|---|
| DialogLast | $0.590 \pm 0.001$ | $0.633 \pm 0.001$ | $0.604 \pm 0.005$ | $0.549 \pm 0.002$ | $0.576 \pm 0.002$ |
| DialogLast+GeoAggregation | $0.636 \pm 0.001$ | $0.603 \pm 0.001$ | $0.649 \pm 0.003$ | $0.645 \pm 0.001$ | $0.648 \pm 0.002$ |
| DialogLast+GeoCoLES | $0.647 \pm 0.002$ | $0.615 \pm 0.002$ | $0.660 \pm 0.006$ | $0.650 \pm 0.002$ | $0.663 \pm 0.003$ |
| DialogLast+GeoTabGPT | $0.654 \pm 0.002$ | $0.631 \pm 0.001$ | $0.673 \pm 0.005$ | $0.652 \pm 0.002$ | $0.662 \pm 0.002$ |
| DialogLast+GeoTabBERT | $0.642 \pm 0.003$ | $0.613 \pm 0.002$ | $0.655 \pm 0.009$ | $0.646 \pm 0.001$ | $0.655 \pm 0.004$ |
| DialogLast+TrxAggregation | $0.788 \pm 0.001$ | $0.749 \pm 0.001$ | $0.826 \pm 0.003$ | $0.772 \pm 0.001$ | $0.805 \pm 0.004$ |
| DialogLast+TrxAggregation+GeoAggregation | $0.787 \pm 0.001$ | $0.746 \pm 0.001$ | $0.826 \pm 0.004$ | $0.773 \pm 0.001$ | $0.804 \pm 0.002$ |
| DialogLast+TrxCoLES | $0.776 \pm 0.002$ | $0.739 \pm 0.003$ | $0.814 \pm 0.004$ | $0.753 \pm 0.002$ | $0.797 \pm 0.003$ |
| DialogLast+TrxCoLES+GeoCoLES | $0.777 \pm 0.002$ | $0.739 \pm 0.004$ | $0.816 \pm 0.004$ | $0.755 \pm 0.003$ | $0.798 \pm 0.002$ |
| DialogLast+TrxTabGPT | $0.805 \pm 0.001$ | $0.775 \pm 0.001$ | $0.842 \pm 0.002$ | $0.778 \pm 0.002$ | $0.823 \pm 0.005$ |
| DialogLast+TrxTabGPT+GeoTabGPT | $0.802 \pm 0.001$ | $0.764 \pm 0.001$ | $0.845 \pm 0.002$ | $0.777 \pm 0.001$ | $0.820 \pm 0.004$ |
| DialogLast+TrxTabBERT | $0.764 \pm 0.009$ | $0.715 \pm 0.019$ | $0.817 \pm 0.007$ | $0.734 \pm 0.007$ | $0.789 \pm 0.008$ |
| DialogLast+TrxTabBERT+GeoTabBERT | $0.765 \pm 0.009$ | $0.714 \pm 0.019$ | $0.819 \pm 0.006$ | $0.738 \pm 0.007$ | $0.789 \pm 0.008$ |
| DialogMean | $0.604 \pm 0.001$ | $0.636 \pm 0.002$ | $0.629 \pm 0.001$ | $0.564 \pm 0.001$ | $0.587 \pm 0.001$ |
| DialogMean+GeoAggregation | $0.642 \pm 0.001$ | $0.605 \pm 0.002$ | $0.658 \pm 0.002$ | $0.650 \pm 0.001$ | $0.654 \pm 0.001$ |
| DialogMean+GeoCoLES | $0.653 \pm 0.001$ | $0.618 \pm 0.004$ | $0.670 \pm 0.002$ | $0.656 \pm 0.001$ | $0.669 \pm 0.001$ |
| DialogMean+GeoTabGPT | $0.661 \pm 0.001$ | $0.633 \pm 0.001$ | $0.681 \pm 0.004$ | $0.657 \pm 0.002$ | $0.671 \pm 0.002$ |
| DialogMean+GeoTabBERT | $0.648 \pm 0.003$ | $0.613 \pm 0.003$ | $0.666 \pm 0.005$ | $0.652 \pm 0.002$ | $0.662 \pm 0.004$ |
| DialogMean+TrxAggregation | $0.788 \pm 0.000$ | $0.749 \pm 0.001$ | $0.825 \pm 0.003$ | $0.773 \pm 0.001$ | $0.804 \pm 0.001$ |
| DialogMean+TrxAggregation+GeoAggregation | $0.787 \pm 0.001$ | $0.746 \pm 0.001$ | $0.825 \pm 0.002$ | $0.773 \pm 0.001$ | $0.804 \pm 0.003$ |
| DialogMean+TrxCoLES | $0.776 \pm 0.002$ | $0.739 \pm 0.003$ | $0.814 \pm 0.004$ | $0.753 \pm 0.002$ | $0.798 \pm 0.003$ |
| DialogMean+TrxCoLES+GeoCoLES | $0.777 \pm 0.002$ | $0.739 \pm 0.002$ | $0.815 \pm 0.003$ | $0.755 \pm 0.003$ | $0.799 \pm 0.002$ |
| DialogMean+TrxTabGPT | $0.805 \pm 0.001$ | $0.775 \pm 0.001$ | $0.843 \pm 0.001$ | $0.778 \pm 0.002$ | $0.824 \pm 0.004$ |
| DialogMean+TrxTabGPT+GeoTabGPT | $0.802 \pm 0.001$ | $0.764 \pm 0.001$ | $0.845 \pm 0.002$ | $0.777 \pm 0.001$ | $0.821 \pm 0.003$ |
| DialogMean+TrxTabBERT | $0.765 \pm 0.009$ | $0.715 \pm 0.019$ | $0.817 \pm 0.006$ | $0.735 \pm 0.006$ | $0.792 \pm 0.007$ |
| DialogMean+TrxTabBERT+GeoTabBERT | $0.766 \pm 0.009$ | $0.714 \pm 0.019$ | $0.819 \pm 0.006$ | $0.738 \pm 0.006$ | $0.792 \pm 0.007$ |
| GeoAggregation | $0.554 \pm 0.001$ | $0.540 \pm 0.001$ | $0.584 \pm 0.002$ | $0.534 \pm 0.001$ | $0.559 \pm 0.001$ |
| GeoCoLES | $0.601 \pm 0.004$ | $0.565 \pm 0.004$ | $0.668 \pm 0.011$ | $0.571 \pm 0.003$ | $0.600 \pm 0.003$ |
| GeoTabGPT | $0.622 \pm 0.001$ | $0.589 \pm 0.001$ | $0.700 \pm 0.008$ | $0.586 \pm 0.003$ | $0.615 \pm 0.004$ |
| GeoTabBERT | $0.596 \pm 0.002$ | $0.566 \pm 0.003$ | $0.663 \pm 0.010$ | $0.570 \pm 0.003$ | $0.585 \pm 0.007$ |
| TrxAggregation | $0.780 \pm 0.005$ | $0.743 \pm 0.001$ | $0.824 \pm 0.001$ | $0.762 \pm 0.001$ | $0.791 \pm 0.017$ |
| TrxAggregation+GeoAggregation | $0.779 \pm 0.004$ | $0.740 \pm 0.001$ | $0.828 \pm 0.002$ | $0.762 \pm 0.001$ | $0.787 \pm 0.013$ |
| TrxCoLES | $0.772 \pm 0.002$ | $0.734 \pm 0.003$ | $0.813 \pm 0.004$ | $0.746 \pm 0.001$ | $0.793 \pm 0.003$ |
| TrxCoLES+GeoCoLES | $0.772 \pm 0.002$ | $0.734 \pm 0.004$ | $0.814 \pm 0.004$ | $0.749 \pm 0.002$ | $0.792 \pm 0.002$ |
| TrxTabGPT | $0.796 \pm 0.000$ | $0.745 \pm 0.001$ | $0.837 \pm 0.004$ | $0.777 \pm 0.001$ | $0.824 \pm 0.005$ |
| TrxTabGPT+GeoTabGPT | $0.796 \pm 0.001$ | $0.751 \pm 0.002$ | $0.843 \pm 0.011$ | $0.774 \pm 0.001$ | $0.816 \pm 0.003$ |
| TrxTabBERT | $0.754 \pm 0.011$ | $0.707 \pm 0.019$ | $0.815 \pm 0.006$ | $0.717 \pm 0.012$ | $0.778 \pm 0.012$ |
| TrxTabBERT+GeoTabBERT | $0.756 \pm 0.011$ | $0.707 \pm 0.019$ | $0.816 \pm 0.005$ | $0.722 \pm 0.010$ | $0.778 \pm 0.012$ |

Table 11: Blending results on public dataset

| methods | mean | target_1 | target_2 | target_3 | target_4 |
|---|---|---|---|---|---|
| DialogLast | $0.586 \pm 0.001$ | $0.602 \pm 0.001$ | $0.622 \pm 0.004$ | $0.554 \pm 0.001$ | $0.567 \pm 0.002$ |
| DialogLast+GeoAggregation | $0.600 \pm 0.001$ | $0.605 \pm 0.000$ | $0.648 \pm 0.004$ | $0.560 \pm 0.001$ | $0.585 \pm 0.002$ |
| DialogLast+GeoCoLES | $0.625 \pm 0.003$ | $0.615 \pm 0.001$ | $0.700 \pm 0.005$ | $0.577 \pm 0.003$ | $0.609 \pm 0.003$ |
| DialogLast+GeoTabGPT | $0.642 \pm 0.002$ | $0.629 \pm 0.001$ | $0.725 \pm 0.007$ | $0.589 \pm 0.002$ | $0.623 \pm 0.002$ |
| DialogLast+GeoTabBERT | $0.629 \pm 0.001$ | $0.616 \pm 0.001$ | $0.709 \pm 0.006$ | $0.577 \pm 0.002$ | $0.613 \pm 0.001$ |
| DialogLast+TrxAggregation | $0.732 \pm 0.002$ | $0.685 \pm 0.001$ | $0.826 \pm 0.004$ | $0.688 \pm 0.001$ | $0.731 \pm 0.003$ |
| DialogLast+TrxAggregation+GeoAggregation | $0.731 \pm 0.002$ | $0.681 \pm 0.001$ | $0.825 \pm 0.005$ | $0.684 \pm 0.001$ | $0.736 \pm 0.003$ |
| DialogLast+TrxCoLES | $0.714 \pm 0.001$ | $0.675 \pm 0.001$ | $0.806 \pm 0.004$ | $0.658 \pm 0.002$ | $0.715 \pm 0.003$ |
| DialogLast+TrxCoLES+GeoCoLES | $0.720 \pm 0.001$ | $0.673 \pm 0.002$ | $0.822 \pm 0.004$ | $0.657 \pm 0.002$ | $0.728 \pm 0.002$ |
| DialogLast+TrxTabGPT | $0.743 \pm 0.002$ | $0.677 \pm 0.001$ | $0.834 \pm 0.004$ | $0.697 \pm 0.001$ | $0.766 \pm 0.003$ |
| DialogLast+TrxTabGPT+GeoTabGPT | $0.753 \pm 0.001$ | $0.684 \pm 0.001$ | $0.848 \pm 0.004$ | $0.700 \pm 0.001$ | $0.779 \pm 0.003$ |
| DialogLast+TrxTabBERT | $0.710 \pm 0.005$ | $0.669 \pm 0.003$ | $0.814 \pm 0.006$ | $0.640 \pm 0.008$ | $0.715 \pm 0.007$ |
| DialogLast+TrxTabBERT+GeoTabBERT | $0.718 \pm 0.004$ | $0.670 \pm 0.003$ | $0.828 \pm 0.005$ | $0.641 \pm 0.008$ | $0.733 \pm 0.006$ |
| DialogMean | $0.595 \pm 0.002$ | $0.600 \pm 0.001$ | $0.633 \pm 0.006$ | $0.566 \pm 0.001$ | $0.580 \pm 0.002$ |
| DialogMean+GeoAggregation | $0.607 \pm 0.001$ | $0.604 \pm 0.000$ | $0.657 \pm 0.004$ | $0.572 \pm 0.000$ | $0.596 \pm 0.002$ |
| DialogMean+GeoCoLES | $0.630 \pm 0.002$ | $0.615 \pm 0.001$ | $0.703 \pm 0.006$ | $0.586 \pm 0.002$ | $0.618 \pm 0.003$ |
| DialogMean+GeoTabGPT | $0.645 \pm 0.002$ | $0.629 \pm 0.001$ | $0.727 \pm 0.010$ | $0.596 \pm 0.001$ | $0.630 \pm 0.001$ |
| DialogMean+GeoTabBERT | $0.634 \pm 0.002$ | $0.616 \pm 0.002$ | $0.713 \pm 0.006$ | $0.586 \pm 0.002$ | $0.621 \pm 0.002$ |
| DialogMean+TrxAggregation | $0.732 \pm 0.001$ | $0.684 \pm 0.001$ | $0.824 \pm 0.001$ | $0.688 \pm 0.001$ | $0.732 \pm 0.002$ |
| DialogMean+TrxAggregation+GeoAggregation | $0.731 \pm 0.001$ | $0.679 \pm 0.000$ | $0.822 \pm 0.002$ | $0.684 \pm 0.001$ | $0.737 \pm 0.002$ |
| DialogMean+TrxCoLES | $0.713 \pm 0.001$ | $0.674 \pm 0.001$ | $0.804 \pm 0.002$ | $0.659 \pm 0.002$ | $0.717 \pm 0.003$ |
| DialogMean+TrxCoLES+GeoCoLES | $0.719 \pm 0.001$ | $0.672 \pm 0.002$ | $0.819 \pm 0.003$ | $0.658 \pm 0.002$ | $0.729 \pm 0.002$ |
| DialogMean+TrxTabGPT | $0.742 \pm 0.001$ | $0.675 \pm 0.001$ | $0.829 \pm 0.004$ | $0.697 \pm 0.001$ | $0.766 \pm 0.003$ |
| DialogMean+TrxTabGPT+GeoTabGPT | $0.751 \pm 0.001$ | $0.682 \pm 0.001$ | $0.845 \pm 0.004$ | $0.700 \pm 0.001$ | $0.779 \pm 0.004$ |
| DialogMean+TrxTabBERT | $0.709 \pm 0.004$ | $0.668 \pm 0.003$ | $0.812 \pm 0.008$ | $0.642 \pm 0.007$ | $0.717 \pm 0.006$ |
| DialogMean+TrxTabBERT+GeoTabBERT | $0.718 \pm 0.003$ | $0.668 \pm 0.003$ | $0.826 \pm 0.006$ | $0.643 \pm 0.007$ | $0.734 \pm 0.006$ |
| GeoAggregation | $0.555 \pm 0.001$ | $0.539 \pm 0.000$ | $0.590 \pm 0.002$ | $0.533 \pm 0.001$ | $0.560 \pm 0.001$ |
| GeoCoLES | $0.598 \pm 0.004$ | $0.568 \pm 0.003$ | $0.663 \pm 0.005$ | $0.568 \pm 0.007$ | $0.593 \pm 0.005$ |
| GeoTabGPT | $0.621 \pm 0.003$ | $0.589 \pm 0.002$ | $0.696 \pm 0.010$ | $0.586 \pm 0.002$ | $0.614 \pm 0.002$ |
| GeoTabBERT | $0.603 \pm 0.002$ | $0.573 \pm 0.003$ | $0.672 \pm 0.007$ | $0.570 \pm 0.004$ | $0.598 \pm 0.004$ |
| TrxAggregation | $0.788 \pm 0.001$ | $0.743 \pm 0.003$ | $0.831 \pm 0.002$ | $0.777 \pm 0.001$ | $0.800 \pm 0.002$ |
| TrxAggregation+GeoAggregation | $0.778 \pm 0.002$ | $0.733 \pm 0.002$ | $0.822 \pm 0.004$ | $0.767 \pm 0.001$ | $0.790 \pm 0.002$ |
| TrxCoLES | $0.774 \pm 0.002$ | $0.734 \pm 0.002$ | $0.812 \pm 0.004$ | $0.759 \pm 0.002$ | $0.790 \pm 0.003$ |
| TrxCoLES+GeoCoLES | $0.775 \pm 0.001$ | $0.730 \pm 0.002$ | $0.827 \pm 0.004$ | $0.751 \pm 0.003$ | $0.792 \pm 0.002$ |
| TrxTabGPT | $0.802 \pm 0.001$ | $0.751 \pm 0.001$ | $0.844 \pm 0.002$ | $0.788 \pm 0.002$ | $0.826 \pm 0.003$ |
| TrxTabGPT+GeoTabGPT | $0.804 \pm 0.001$ | $0.748 \pm 0.001$ | $0.854 \pm 0.002$ | $0.784 \pm 0.001$ | $0.829 \pm 0.003$ |
| TrxTabBERT | $0.762 \pm 0.004$ | $0.717 \pm 0.006$ | $0.819 \pm 0.004$ | $0.734 \pm 0.006$ | $0.778 \pm 0.006$ |
| TrxTabBERT+GeoTabBERT | $0.766 \pm 0.004$ | $0.717 \pm 0.006$ | $0.831 \pm 0.005$ | $0.729 \pm 0.007$ | $0.786 \pm 0.005$ |

Table 12: Late Fusion results on public dataset

| methods | mean | target_1 | target_2 | target_3 | target_4 |
|---|---|---|---|---|---|
| DialogLast | $0.586 \pm 0.001$ | $0.602 \pm 0.001$ | $0.622 \pm 0.004$ | $0.554 \pm 0.001$ | $0.567 \pm 0.002$ |
| DialogLast+GeoAggregation | $0.646 \pm 0.001$ | $0.614 \pm 0.001$ | $0.659 \pm 0.003$ | $0.654 \pm 0.001$ | $0.657 \pm 0.001$ |
| DialogLast+GeoCoLES | $0.660 \pm 0.001$ | $0.633 \pm 0.002$ | $0.675 \pm 0.004$ | $0.661 \pm 0.001$ | $0.671 \pm 0.003$ |
| DialogLast+GeoTabGPT | $0.668 \pm 0.001$ | $0.645 \pm 0.001$ | $0.690 \pm 0.004$ | $0.662 \pm 0.001$ | $0.674 \pm 0.002$ |
| DialogLast+GeoTabBERT | $0.662 \pm 0.001$ | $0.633 \pm 0.002$ | $0.680 \pm 0.003$ | $0.662 \pm 0.001$ | $0.675 \pm 0.002$ |
| DialogLast+TrxAggregation | $0.792 \pm 0.002$ | $0.752 \pm 0.001$ | $0.829 \pm 0.001$ | $0.780 \pm 0.001$ | $0.805 \pm 0.006$ |
| DialogLast+TrxAggregation+GeoAggregation | $0.791 \pm 0.001$ | $0.750 \pm 0.001$ | $0.829 \pm 0.005$ | $0.782 \pm 0.001$ | $0.803 \pm 0.001$ |
| DialogLast+TrxCoLES | $0.782 \pm 0.001$ | $0.746 \pm 0.003$ | $0.814 \pm 0.003$ | $0.765 \pm 0.002$ | $0.802 \pm 0.003$ |
| DialogLast+TrxCoLES+GeoCoLES | $0.783 \pm 0.001$ | $0.745 \pm 0.004$ | $0.819 \pm 0.003$ | $0.767 \pm 0.001$ | $0.803 \pm 0.002$ |
| DialogLast+TrxTabGPT | $0.810 \pm 0.001$ | $0.779 \pm 0.001$ | $0.846 \pm 0.003$ | $0.789 \pm 0.002$ | $0.827 \pm 0.004$ |
| DialogLast+TrxTabGPT+GeoTabGPT | $0.808 \pm 0.001$ | $0.770 \pm 0.001$ | $0.849 \pm 0.003$ | $0.790 \pm 0.001$ | $0.824 \pm 0.004$ |
| DialogLast+TrxTabBERT | $0.773 \pm 0.002$ | $0.730 \pm 0.006$ | $0.822 \pm 0.005$ | $0.749 \pm 0.004$ | $0.792 \pm 0.003$ |
| DialogLast+TrxTabBERT+GeoTabBERT | $0.776 \pm 0.003$ | $0.729 \pm 0.006$ | $0.827 \pm 0.004$ | $0.752 \pm 0.004$ | $0.794 \pm 0.003$ |
| DialogMean | $0.595 \pm 0.002$ | $0.600 \pm 0.001$ | $0.633 \pm 0.006$ | $0.566 \pm 0.001$ | $0.580 \pm 0.002$ |
| DialogMean+GeoAggregation | $0.649 \pm 0.001$ | $0.614 \pm 0.001$ | $0.665 \pm 0.002$ | $0.656 \pm 0.001$ | $0.662 \pm 0.001$ |
| DialogMean+GeoCoLES | $0.663 \pm 0.001$ | $0.632 \pm 0.002$ | $0.680 \pm 0.002$ | $0.663 \pm 0.000$ | $0.675 \pm 0.002$ |
| DialogMean+GeoTabGPT | $0.670 \pm 0.001$ | $0.645 \pm 0.000$ | $0.694 \pm 0.004$ | $0.664 \pm 0.001$ | $0.678 \pm 0.001$ |
| DialogMean+GeoTabBERT | $0.664 \pm 0.001$ | $0.633 \pm 0.001$ | $0.682 \pm 0.002$ | $0.664 \pm 0.001$ | $0.678 \pm 0.001$ |
| DialogMean+TrxAggregation | $0.792 \pm 0.002$ | $0.752 \pm 0.002$ | $0.828 \pm 0.002$ | $0.781 \pm 0.001$ | $0.807 \pm 0.006$ |
| DialogMean+TrxAggregation+GeoAggregation | $0.792 \pm 0.002$ | $0.750 \pm 0.002$ | $0.829 \pm 0.002$ | $0.782 \pm 0.001$ | $0.807 \pm 0.005$ |
| DialogMean+TrxCoLES | $0.781 \pm 0.001$ | $0.745 \pm 0.003$ | $0.814 \pm 0.002$ | $0.765 \pm 0.002$ | $0.802 \pm 0.003$ |
| DialogMean+TrxCoLES+GeoCoLES | $0.783 \pm 0.001$ | $0.744 \pm 0.004$ | $0.819 \pm 0.003$ | $0.767 \pm 0.002$ | $0.802 \pm 0.002$ |
| DialogMean+TrxTabGPT | $0.810 \pm 0.002$ | $0.779 \pm 0.001$ | $0.847 \pm 0.004$ | $0.789 \pm 0.001$ | $0.828 \pm 0.003$ |
| DialogMean+TrxTabGPT+GeoTabGPT | $0.808 \pm 0.001$ | $0.770 \pm 0.001$ | $0.848 \pm 0.003$ | $0.790 \pm 0.001$ | $0.826 \pm 0.003$ |
| DialogMean+TrxTabBERT | $0.773 \pm 0.003$ | $0.730 \pm 0.006$ | $0.822 \pm 0.004$ | $0.749 \pm 0.004$ | $0.791 \pm 0.003$ |
| DialogMean+TrxTabBERT+GeoTabBERT | $0.775 \pm 0.003$ | $0.728 \pm 0.005$ | $0.827 \pm 0.003$ | $0.752 \pm 0.004$ | $0.794 \pm 0.004$ |
| GeoAggregation | $0.555 \pm 0.001$ | $0.539 \pm 0.000$ | $0.590 \pm 0.002$ | $0.533 \pm 0.001$ | $0.560 \pm 0.001$ |
| GeoCoLES | $0.598 \pm 0.004$ | $0.568 \pm 0.003$ | $0.663 \pm 0.005$ | $0.568 \pm 0.007$ | $0.593 \pm 0.005$ |
| GeoTabGPT | $0.621 \pm 0.003$ | $0.589 \pm 0.002$ | $0.696 \pm 0.010$ | $0.586 \pm 0.002$ | $0.614 \pm 0.002$ |
| GeoTabBERT | $0.603 \pm 0.002$ | $0.573 \pm 0.003$ | $0.672 \pm 0.007$ | $0.570 \pm 0.004$ | $0.598 \pm 0.004$ |
| TrxAggregation | $0.783 \pm 0.002$ | $0.741 \pm 0.003$ | $0.828 \pm 0.003$ | $0.770 \pm 0.004$ | $0.792 \pm 0.007$ |
| TrxAggregation+GeoAggregation | $0.783 \pm 0.002$ | $0.740 \pm 0.002$ | $0.829 \pm 0.003$ | $0.771 \pm 0.001$ | $0.792 \pm 0.011$ |
| TrxCoLES | $0.773 \pm 0.002$ | $0.734 \pm 0.002$ | $0.812 \pm 0.004$ | $0.758 \pm 0.002$ | $0.790 \pm 0.003$ |
| TrxCoLES+GeoCoLES | $0.775 \pm 0.002$ | $0.734 \pm 0.002$ | $0.815 \pm 0.004$ | $0.760 \pm 0.002$ | $0.789 \pm 0.003$ |
| TrxTabGPT | $0.802 \pm 0.001$ | $0.751 \pm 0.001$ | $0.844 \pm 0.002$ | $0.787 \pm 0.001$ | $0.825 \pm 0.003$ |
| TrxTabGPT+GeoTabGPT | $0.800 \pm 0.001$ | $0.752 \pm 0.001$ | $0.846 \pm 0.005$ | $0.785 \pm 0.002$ | $0.817 \pm 0.006$ |
| TrxTabBERT | $0.762 \pm 0.004$ | $0.717 \pm 0.006$ | $0.819 \pm 0.004$ | $0.734 \pm 0.006$ | $0.777 \pm 0.006$ |
| TrxTabBERT+GeoTabBERT | $0.764 \pm 0.004$ | $0.716 \pm 0.006$ | $0.823 \pm 0.004$ | $0.737 \pm 0.006$ | $0.780 \pm 0.005$ |

