# OpenReview forum: "Multimodal Banking Dataset: Understanding Client Needs through Event Sequences"
_ICLR.cc/2025/Conference — Submitted to ICLR 2025_

### Official Review · Reviewer_PGwK · 2024-10-28

**Soundness:** 2
**Presentation:** 3
**Contribution:** 2
**Rating:** 5
**Confidence:** 3

**Summary:**

The paper introduces a new multimodal dataset, MBD, from over 2 million corporate clients, including bank transactions, geo-locations, technical support dialogues, and monthly aggregated purchases of four banking products. The authors also present a benchmark for evaluating models on this dataset and two other financial datasets, focusing on tasks like purchase prediction and multimodal matching.

**Strengths:**

- Large-scale dataset: The MBD dataset is the largest of its kind, offering a significant amount of data for research purposes.

- Multimodality: The dataset incorporates various data modalities, providing a more comprehensive view of client behavior.

- Practical tasks: The benchmark focuses on practically relevant tasks, such as purchase prediction, which can be useful for real-world applications.

- Anonymization: The authors have taken steps to anonymize the data, protecting client privacy.

**Weaknesses:**

- Lack of novelty: The paper primarily focuses on introducing a dataset and benchmark. For ICLR submissions, I'd see more emphasis on novel methods or algorithms. In addition, a careful evaluation of previous models (for example, the MMBench https://arxiv.org/abs/2307.06281) will bring more novelty to the community.

- Missing comparison with LLMs: The paper lacks a comparison with more recent and powerful language models like GPT-4 or BloombergGPT, which have shown strong performance in various financial NLP tasks. For example, can we use prompt engineering to guide the GPT4o to process the transactions and geo-locations.

- Unclear practical impact of the proposed metric: While the paper mentions that the benchmark can lead to financial benefits, I am curious if real world users care recalls/AUCs, or there are better metrics that map to financial success.

**Questions:**

-  The paper could be strengthened by exploring more advanced multimodal fusion techniques beyond late fusion.
-  A more detailed analysis of the anonymization process and its potential impact on model performance would be beneficial.
-  The authors could consider expanding the benchmark to include other relevant tasks, such as risk assessment or fraud detection.

Overall, while the MBD dataset and benchmark are valuable contributions, the paper needs significant revisions to address the lack of novelty and provide a more convincing argument for the practical impact of their work.

---

> ### Author Response · Authors · 2024-11-21
>
> We sincerely thank the reviewer for their valuable feedback and have addressed the points raised as follows:
>
> **W1. Lack of novelty.**
>
> We addressed this in general response (item 5). Additionally, we would like to highlight that our benchmark has been carefully designed to encompass a wide range of state-of-the-art approaches relevant to our domain, providing a robust foundation for comprehensive evaluation and advancing research. To further enhance our work, we plan to integrate several advanced methods, including multimodal fusion techniques (general section, item 2) and cutting-edge LLMs (general response, item 1), which are discussed in greater detail later in the paper.
>
> **W2. Comparison with LLMs.**
>
> We addressed this in the general response (item 1).
>
> **W3. Unclear practical impact.**
>
> Thanks for this very important question. In real-world business scenarios, the effectiveness of campaigns is ultimately measured by revenue. However, it is impossible to estimate the quality of every ML model using expensive A/B tests. Hence, it is typical to compare models for prediction success of campaigning by using ROC AUC as a proxy metric, and choose only the top model for further A/B study. This approach has been rigorously validated through real-world A/B tests and is confirmed as the core metric for these tasks by our colleagues from one of the largest banks in the world, underscoring its practical and financial value. There exist several explanations to support the choice of AUC for campaigning. For example, as it is stated in the following paper from WSDM (https://dl.acm.org/doi/abs/10.1145/2124295.2124353), "Selecting a rank at which to evaluate precision such that it would be suitable for all campaigns, is not possible. Instead, we use AUC since it combines the prediction performance over all ranks into a single number".  Its is also known for a long time (KDD’01, https://dl.acm.org/doi/abs/10.1145/502512.502581), when comparing two models by their ROC curves, we’re actually comparing their RNR (Non-Response Ratio) at all possible cutoff points simultaneously.
>
> **Q1. Advanced multimodal fusion techniques.**
>
> We addressed this in the general response (item 2).
>
> **Q2. Detailed analysis of the anonymization process.**
>
> Thank you for your valuable feedback. We have addressed this topic in the general response (item 3). Could you kindly share any other suggestions about specific experiments that should be carried out?
>
> **Q3. Expand benchmark tasks.**
>
> Thanks for your suggestion the inclusion of risk assessment and fraud detection tasks in the benchmark. We have addressed this topic in the general response (item 4). Briefly speaking, we have some plans about extending the dataset, but any new benchmark with financial data requires huge efforts, which is the reason why our dataset is actually the first large-scale multimodal banking dataset. Anyway, it is necessary to highlight that our internal experiments with private data demonstrate that our current set of tasks introduced in the MBD dataset is enough to establish a good ranking of methods even for other tasks of risk assessment. Indeed, campaigning tasks are based on predicting the future behavior of a client, which in some sense is relevant to risks and churn.

---

> > ### Author Response · Authors · 2024-11-28
> >
> > We sincerely thank the reviewer for their insightful feedback, which has been instrumental in improving the quality of our work. In particular, we have thoroughly addressed your comments by incorporating results on advanced multimodal fusion techniques and providing a detailed comparison of Late Fusion and Early Fusion approaches.
> >
> > Furthermore, we would like to emphasize that new results comparing our methods with LLM-based approaches will be incorporated into the final version of the paper after recalculating them for the full validation set described in the article. Due to the complexity of these experiments, they require significant computational time to complete.

---

> > > ### Comment · Reviewer_PGwK · 2024-12-01
> > >
> > > I would like to thank the reviewers for the replies. After reading the responses together with other reviewers' comments, I'd like to keep my previous rating.

---

### Official Review · Reviewer_hmLd · 2024-11-02

**Soundness:** 1
**Presentation:** 1
**Contribution:** 1
**Rating:** 3
**Confidence:** 5

**Summary:**

This paper introduces a large-scale publicly available multimodal banking dataset (Multimodal Banking Dataset, MBD). The MBD contains data from over 2 million banking customers, covering four different modalities: 9.5 million banking transaction records, 1 billion geographic location data points, 5 million customer interactions with technical support embedded dialogues, and banking product purchase activity over a 12-month period. This dataset provides researchers with a rich resource for analyzing customer behavior dynamics and contributes to the development of large-scale multimodal event sequence algorithms in the future.

**Strengths:**

1.A large-scale multimodal banking dataset, MBD, is provided. This dataset contains anonymized banking transactions, geographic locations, and technical support dialogues, which contributes to the development of large-scale sequential event tasks in the future.
2.The dataset addresses privacy concerns through effective data anonymization, ensuring that the algorithm's performance is not significantly compromised.
3.The dataset and experimental code are publicly available, promoting transparency and reproducibility in research.

**Weaknesses:**

1.The dataset's multimodal data includes banking transaction records, geographic locations, dialogue embeddings, and banking product purchase history. However, it appears that many of these modalities are essentially text-based. This differs from typical multimodal datasets, which include modalities such as video, audio, and text.

2.The main contribution of the paper lies in the introduction of a large-scale dataset, but it lacks innovative methods for addressing related tasks. Additionally, the experimental section presents insufficient comparisons with current state-of-the-art methods. Overall, the paper's innovativeness needs improvement. I suggest that the authors include more experiments involving fully supervised methods [1][2]. Additionally, it would be beneficial to propose a simple and practical innovative approach based on their dataset.
[1]Learning Deep Time-index Models for Time Series Forecasting. [2]Crossformer: Transformer Utilizing Cross-Dimension Dependency for Multivariate Time Series Forecasting

3.This paper contains numerous grammatical and tense issues. Additionally, the expression in the paper is not sufficiently clear, making it difficult for readers to accurately understand certain points and arguments. For example, on page three, phrases like “we selected” and “the dataset was collected”; on page five, “the coordinates were”; line 225 includes “we concentrated,” “whether it was,” and “random noise was added.” Additionally, on line 295, “we propose a downstream task — multimodal matching” (Zong et al., 2023).

**Questions:**

1. Do you have any specific method to model the different text modalities?
2.The paper contains numerous tense and grammatical errors, which do not meet the standards expected for ICLR.

---

> ### Author Response · Authors · 2024-11-21
>
> We thank the reviewer for thorough feedback and address the raised concerns as outlined below:
>
> **W1. Non-traditional text-based modalities.**
>
> Thank you for this valuable comment. The primary objective of our dataset is to address key tasks in the banking industry such as fraud detection, credit risk assessment, customer segmentation, personalized recommendations and campaigning.  As highlighted in Section 3: Proposed Dataset, these tasks heavily rely on large-scale event-based data — including transactions, geolocations, and product purchase histories—rather than traditional modalities like images or audio, which have limited applicability in this domain.
>
> As outlined in Section 3.1, our data sources encode diverse types of information: transactions provide numerical and categorical insights into financial behavior, geolocations capture spatial movement patterns, and dialogue embeddings deliver semantic information from customer interactions. Based on our experience with data from one of the world’s largest banks, images, videos, and text (e.g., transcriptions) play a marginal role, while event/tabular data constitutes the vast majority of data processed. Thus, we strongly believe that the difference from most existing multimodal datasets focused on video, audio, and textual modalities is essentially the main strength of our dataset as it lets the researchers focus on real industrial use-cases.
>
> Contrary to the assumption that our data is text-based, it lacks vocabularies and linguistic structures essential for text processing. While sequential in nature, it differs fundamentally from text, images, or audio, requiring tailored, modality-specific methods. Furthermore, our experiments with general LLM-based methods on this dataset revealed that domain-specific event sequence models consistently outperformed them (see general response, item 1).
>
> Given these distinctions, existing multimodal models developed for audiovisual or text processing cannot be directly applied to our dataset. By centering on business-critical tasks, our dataset addresses a significant gap and provides a foundation for advancing innovative multimodal methods tailored to our domain.
>
> **W2. Lack of innovative methods.**
>
> We addressed this in the general response (item 5).
>
> **W3. Numerous tense and grammatical errors.**
>
> During preparation of our paper, we utilized a professional editing service (commercial version of Grammarly) to review the manuscript and improve its clarity and linguistic precision. We are very sorry to hear that our tense usage makes it difficult to accurately understand our certain points and arguments. As we were not aware of standards expected for ICLR, we used Nature’s advice (Section “Using the right tense” in https://www.nature.com/scitable/topicpage/effective-writing-13815989/). For instance, regarding your examples of different tenses, we tried to distinguish between the narrative in the paper in present simple (“we propose”) and some efforts in gathering it before writing the paper (“the dataset was collected”).
>
> Finally, could you provide more details about your comment, “Additionally, the expression in the paper is not sufficiently clear, making it difficult for readers to accurately understand certain points and arguments”? The given examples are just a mixture of present simple and past simple tenses, so it is unclear what difficulties they can bring to readers in terms of understanding our points or arguments. We would be kindly obliged if you could mention sentences or phrases that look weird to you.

---

> > ### Comment · Reviewer_hmLd · 2024-11-26
> >
> > Based on the author's response, I will maintain my original assessment. Regardless of the structure of the text, they all belong to the same type of text and can be processed using the same framework at present, as the author has not proposed a new learning framework.

---

> > > ### Author Response · Authors · 2024-11-28
> > >
> > > Thank you for your feedback. In response to your comments, we have carefully revised the paper to incorporate results on advanced multimodal fusion techniques and provide a thorough comparison of Late Fusion and Early Fusion approaches.
> > > We have also corrected inconsistencies related to tense usage throughout the paper and have made significant efforts to improve its overall clarity and readability. We hope these changes enhance the quality of presentation and comprehensibility of our work.

---

### Official Review · Reviewer_e7p1 · 2024-11-03

**Soundness:** 4
**Presentation:** 2
**Contribution:** 4
**Rating:** 8
**Confidence:** 4

**Summary:**

This is not a research contribution per se but it is a dataset paper.

The paper presents the first large-scale multimodal banking dataset for the user community. It offers a new dataset that has millions of users and millions of transactions that have been suitably anonymized. Moreover, the authors have provided baselines for a few standard tasks. The dataset will be released in order to spur research in the use of machine learning for banking applications.

**Strengths:**

(1) This will be first and the largest multimodal banking dataset that will be released. This can potentially be tremendously useful to the research community.

(2) The baseline methods and benchmark data for a few problems outlined will also be immensely useful to the research community.

**Weaknesses:**

(1) The details of the data are somewhat sparse. More details of each type of data will be useful to the reader. Perhaps this article may be useful for improving this aspect of the exposition in the paper:

https://cacm.acm.org/research/datasheets-for-datasets/

(2)  Can some details of the anonymization be provided without compromising on the privacy of the customers? That can help estimate the errors of any model developed using this data.

(3) The data has been collected during the pandemic period? Will it have any effect on any of the conclusions drawn using the data? For examples, could this lead to systemic under-estimation or over-estimation of any phenomenon? Ideally, it would be useful to have another data-set which is outside of the pandemic period - perhaps the second version of this data set? This can serve as a basis for many natural experiments.

**Questions:**

(1) Will the dataset be openly downloadable or will the authors be controlling the access? Openly downloadable option is obviously preferable.
(2) Will the source code of the bench-marking studies be openly available?
(3) it will be useful to the community if the set of relevant real-world problems could be articulated that can inspire researchers to work on this dataset. Especially, in order to attract young researchers into the field.

**Details Of Ethics Concerns:**

It is hoped that the permission of the appropriate banking regulator (in the jurisdiction of the authors) has been taken in order to release the dataset. The authors, the authors' institution as well as researchers in the field should not be exposed to any litigation risk by a customer or a group of customers.

---

> ### Author Response · Authors · 2024-11-21
>
> We are grateful for the high appreciation of our work and valuable feedback! Here, we provide a detailed response to the raised points.
>
> **W1. Details of the data.**
>
> Thank you for your feedback. Below, we address points from the datasheets-for-datasets framework with references to specific sections of our paper:
>
> Motivation
>
> See Section 1. Introduction. Our primary motivation for sharing proprietary data is to enable and encourage research teams to develop innovative methods for handling multimodal event stream data.
>
> Composition
>
> Appendix A includes data samples and detailed feature statistics for each modality, while Section 3. Proposed dataset provides key statistics summarizing the dataset.
>
> Collection Process
>
> Permissible details about the dataset's collection process are discussed in Section 3. Proposed Dataset.
>
> Preprocessing, Cleaning, and Labeling
>
> Permissible details about preprocessing, cleaning, and anonymization methods are covered in Section 3.2. Data Anonymization.
>
> Uses
>
> The dataset supports tasks like predicting product purchases, multimodal matching, and next-event prediction. While anonymization might slightly affect data distribution, results strongly correlate with those from the proprietary dataset, ensuring its reliability(See Figure 2).
>
> Distributions and Maintenance
>
> Refer to answers Q1-Q2 for information about data distribution and long-term maintenance.
>
> Together, these details provide a comprehensive overview of our dataset’s documentation, addressing all key aspects. Let us know if further clarifications are needed.
>
> **W2. Details of the anonymization.**
>
> We addressed this in the general response (item 3).
>
> **W3. Pandemic period.**
>
> Thanks for this question! We would like to clarify that our dataset includes data not only from 2021 but also from 2022 (see Section 3. Proposed dataset), providing a broader temporal scope that captures patterns beyond the peak of the pandemic. This historical context helps mitigate concerns about systematic underestimation or overestimation of phenomena and ensures that the dataset reflects more generalizable trends. Moreover, we’re going to maintain our dataset and extend it with additional data when possible.
>
> **Q1-Q2. Public availability**
>
> Thanks, it is an important question. We will provide the publicly available link to the dataset and code in the paper after double-blind review. We confirm that the dataset and the source code for the benchmarking studies will be fully accessible to everyone with open access. Furthermore, the authors will maintain the dataset and code to ensure long-term usability, reliability, and relevance for the research community.
>
> **Q3. Inspiring real-world applications.**
>
> Thank you for the suggestion. We agree that defining relevant real-world problems can inspire researchers and attract new talent. The MBD dataset is currently designed for tasks like campaigning and multimodal matching, but it also provides a foundation for developing methods that could be extended to other business tasks, such as customer segmentation, risk assessment, fraud detection, and client retention. As outlined in Section 4.1. Datasets and Downstream Tasks,  campaigning, in particular, is a critical task, as its success directly impacts the volume of products sold by the bank and its overall profitability. Additionally, high-quality recommendations are essential for enhancing the customer experience, making this task not only important for business but also for customer satisfaction. While these tasks are not yet included, we plan to continue publishing similar datasets in the future and aim to introduce one that incorporates such challenges. We will also revise the text of the article to clarify the opportunities provided by the current version of the dataset.

---

> > ### Comment · Reviewer_e7p1 · 2024-11-23
> >
> > Thank you for your response - it is helpful.

---

### Official Review · Reviewer_YTbu · 2024-11-03

**Soundness:** 3
**Presentation:** 3
**Contribution:** 3
**Rating:** 6
**Confidence:** 3

**Summary:**

1. The paper introduces a new dataset, Multimodal Banking Dataset which integrates multiple modalities for over 2 million corporate clients.
2. The authors highlight the potential applications of this dataset for campaign planning and client behavior analysis, using multimodal benchmarks to demonstrate its value along with baseline model implementation.

**Strengths:**

1. The work releases a first large scale banking dataset for public availability for financial applications.
2. The authors present a good benchmark comparing unimodal and multimodal methods across various predictive tasks. The experimental protocol and metrics are clearly laid out as well.

**Weaknesses:**

1. The authors do not explore or discuss advanced multimodal sequence models or advanced fusion techniques' cross-attention mechanisms as they can better capture interactions across modalities. They mention it at the end as a scope for future work.
2. Though authors discuss using AUC ROC as their metric for mitigating label imbalance issues for example in their campaigning downstream task, they do not discuss or incorporate any additional techniques for handling the label imbalance.
3. Details about anonymization techniques applied are mentioned in the paper but it lacks quantitative evaluation of the impact of these techniques on temporal dependencies within the data.

**Questions:**

1. Is there a plan to expand the set of downstream tasks in future work? Highlighting a larger application list can increase the dataset's appeal across different financial research areas.

Please look into the weakness section for other questions.

---

> ### Author Response · Authors · 2024-11-21
>
> We appreciate the reviewer's detailed feedback and have addressed the points raised as follows:
>
> **W1. Advanced multimodal fusion techniques and sequence models.**
>
> Thank you for your insightful comment. Advanced multimodal fusion techniques are addressed in the general response (item 2). Regarding multimodal sequence models, Sections 4.2 and 5 provide detailed descriptions and experimental results for methods such as TabGPT and TabBERT. These Transformer-based architectures leverage the state-of-the-art capabilities of the Transformer framework, widely recognized for its versatility across diverse data modalities, highlighting its effectiveness for multimodal sequence analysis.
>
> **W2. Additional techniques for handling the label imbalance.**
>
> We acknowledge the importance of addressing label imbalance more comprehensively. Currently, we are conducting experiments with techniques such as over-sampling, under-sampling, and stratified batching to account for the target distribution in supervised methods. These approaches aim to improve the robustness of our models and provide more reliable results in the presence of imbalanced labels. We will include these findings in our updated analysis and revised paper till the end of the discussion phase.
>
> **W3. Anonymization.**
>
> For sure, any anonymization of the timestamps (including ours) may remove some specific temporal patterns in data, but our experiment proves that our anonymization techniques still save all significant information for introduced downstream tasks. We provide more details in our general response (item 3). Could you kindly clarify what additional experiments should be carried out to find the most useful temporal dependencies?
>
> **Q1. Expand the set of downstream tasks in future work.**
> We addressed this in the general response above (item 4).

---

> > ### Comment · Reviewer_YTbu · 2024-11-25
> >
> > 1. Thanks for considering my comments on advanced multimodal fusion techniques and sequence models and including your findings on the same.
> > 2. Updating your analysis on how to improve the robustness of your models by addressing label imbalance will indeed be helpful.
> > 3. I understand that your results prove that anonymization has minimal impact on the campaign targeting task. Your mentioned future tasks, for example, churn prediction and fraud detection might rely on fine-grained temporal dependencies. Would the current anonymization still preserve the necessary information is the question.
> > 4. Thanks for expanding on the set of downstream tasks.

---

> > > ### Author Response · Authors · 2024-11-28
> > >
> > > Thank you for highlighting the potential impact of anonymization on tasks sensitive to fine-grained temporal dependencies. When new targets are introduced, we will reassess the dataset by comparing the performance on anonymized and original data to identify any effects.
> > >
> > > If anonymization is found to compromise temporal dependencies, we will adjust the method to better preserve essential information while maintaining strict privacy. Our current approach employs random salts in hashed IDs, ensuring robust security without retaining the salts. Consequently, updates such as adding new targets or making other modifications result in a new version of the dataset. This feature, driven by stringent security requirements, allows us the flexibility to refine the anonymization process as needed.
> > >
> > > For the updated dataset, we will recompute benchmarks for existing tasks and assess performance on new tasks and models. This approach ensures comprehensive evaluation and continuity across dataset versions.
> > >
> > > To address label imbalance, we experimented with various techniques, including random undersampling, oversampling, and class balancing within gradient boosting on CoLES embeddings. Additionally, stratified batching was applied to better align the target distribution for Supervised RNN. The results of these experiments are summarized in the table below:
> > > | Method                                  |ROC-AUC |
> > > |---------------------------------------- |--------|
> > > | **Supervised RNN**         		  |  0.819 |
> > > | **Supervised RNN (stratified batching)**|  0.819 |
> > > | **CoLES**                               |  0.773 |
> > > | **CoLES (undersampling)**               |  0.772 |
> > > | **CoLES (oversampling)**                |  0.774 |
> > > | **CoLES (class balanced)**              |  0.774 |
> > >
> > > Stratified batching maintains a high ROC-AUC of 0.819 for Supervised RNN, demonstrating its robustness to label imbalance. For CoLES embeddings, balancing techniques yield minimal performance variations, with ROC-AUC ranging from 0.772 to 0.774, indicating limited sensitivity to label imbalance.
> > >
> > > We have included these findings in the updated version of the paper, specifically in Appendix B. Once again, we appreciate the reviewer’s feedback, which allowed us to strengthen our analysis and provide additional insights.

---

### Author Response · Authors · 2024-11-21

Thank you very much for your comments, which allowed us to address the shortcomings and refine the presentation of the paper.  Below we provide additional experimental results to be included in the paper. Some experiments are still in progress and will be finalized soon. We believe these updates significantly strengthen our work. The key updates are summarized below:

**1. LLM-based methods for our dataset.**

Initially, we experimented with several methods for integrating transactional data into LLMs, but their performance lagged significantly behind specialized baselines, so we did not plan to include these results in the paper. Using a relevant article from NeurIPS’23 (https://openreview.net/forum?id=md68e8iZK1), we conducted preliminary experiments with one of the leading LLMs, which is known to be one of the best models for the language of our dataset. We focused on transactional data from our MBD dataset. Transactions were transformed into text format, processed by the LLM to generate embeddings, and used as input for a gradient-boosting model.
To speed up the evaluation during the rebuttal period, we considered predictions for the last month only. We examined two kinds of implementation: 1) the pre-trained LLM; and 2) a fine-tuned version of this LLM (setting similar to BloombergGPT). Zero-shot text-pretrained LLM showed considerably lower quality compared to CoLES, while domain fine-tuning improved performance of LLM, though it still did not surpass CoLES. An alternative method, ESQA (https://arxiv.org/abs/2407.12833), which employs a specialized encoder to structure transactional data, also failed to outperform baseline.

|                | target_1      | target_2      | target_3      | target_4      |
|----------------|---------------|---------------|---------------|---------------|
| CoLES + LLM FT | 0.708 ± 0.008 | 0.799 ± 0.030 | 0.777 ± 0.010 | 0.881 ± 0.004 |
| CoLES          | 0.690 ± 0.012 | 0.794 ± 0.036 | 0.756 ± 0.015 | 0.878 ± 0.004 |
| LLM FT         | 0.601 ± 0.023 | 0.676 ± 0.032 | 0.706 ± 0.017 | 0.819 ± 0.007 |
| LLM zero-shot  | 0.552 ± 0.019 | 0.642 ± 0.010 | 0.699 ± 0.012 | 0.770 ± 0.013 |
| ESQA           | 0.592 ± 0.018 | 0.704 ± 0.028 | 0.558 ± 0.013 | 0.777 ± 0.014 |

Moreover, we discovered that LLM methods can extract new valuable features from transactional data, as confirmed by the improved downstream task performance (up to 2% comparing to CoLES) when LLM embeddings were concatenated with CoLES embeddings (“CoLES + LLM FT” in this Table). This is a promising result, indicating the potential of LLMs for transactional data analysis. We will include these new results in the paper after recalculating them for the whole validation set described in the article, what requires additional time. Our preliminary estimates show that we need 2 weeks to reproduce the complete validation pipeline using LLMs from this Table.

**2. Advanced multimodal fusion techniques.**

To demonstrate the possibility of using more advanced multimodal fusion techniques, we conducted additional experiments of early fusion methods using the cross-attention mechanism, one of the foundations of modern multimodal models mentioned by the reviewers ([1] https://arxiv.org/pdf/2206.06488) and **Crossformer** ([2]  https://openreview.net/pdf?id=vSVLM2j9eie ).

In the following table, we compared the performance of **CrossTransformer** (early fusion, supervised, transformer) with the results presented in our paper for the **Supervised (RNN)** (late fusion, supervised, RNN).

| Method                            | Modalities     |  ROC-AUC |
|----------------------------- |--------------- |--------------|
| **Supervised (RNN)**       | Trx + Dialog  |  0.821        |
| **CrossTransformer**       | Trx + Dialog  |  0.821        |
| **Supervised (RNN)**       | Trx + Geo     |  0.819        |
| **CrossTransformer**       | Trx + Geo     |  0.815        |

The analysis revealed that **CrossTransformer** performs similarly to or slightly worse than the **Supervised (RNN)** method. To examine the impact of Transformer architectures on the performance of multimodal data processing, we are conducting additional experiments with the **Supervised (Transformer)** model (late fusion, supervised, transformer), built on the same principles as **Supervised (RNN)**. We will include the results of a comprehensive comparison of different groups of methods (late fusion and early fusion) into the paper to provide a complete and thorough comparison of multimodal methods.

---

### Author Response · Authors · 2024-11-21

**3. Anonymization.**

Any anonymization (including ours) may remove some specific patterns in data and, hence, impact performance on some downstream tasks, but our experiment proves that our anonymization techniques still save all significant information for introduced downstream tasks. In Subsection 3.2 of our paper, we provided all permissible details of the anonymization process applied to our data. Additionally, Subsection 5.2.1 discusses the impact of anonymization by comparing model performance on the public dataset to that on the private (non-anonymized) dataset version. Figure 2 demonstrates a strong correlation between the results, indicating that the anonymization process has minimal impact on model performance for the downstream task of campaigning. This consistency underscores the reliability of our dataset for benchmarking and advancing research in multimodal event sequence analysis. Additional experimental comparison of results on our MDB and private datasets are available in Appendix B (Tables 6-9). The ranking of methods remains consistent despite differences in absolute metric values. As shown in Table 7, TrxTabGPT (ROC-AUC 0.796) and TrxAggregation (ROC-AUC 0.780) achieve the best performance on the private dataset. Similarly, TrxTabGPT leads on the public dataset with a ROC-AUC of 0.802 (Table 9). Incorporating geolocation and dialogue modalities further improves results, with DialogLast+TrxTabGPT+GeoTabGPT attaining the highest ROC-AUC on both datasets—0.802 on the private dataset (Table 7) and 0.808 on the public dataset (Table 9). Overall, multimodal approaches utilizing TabGPT or Aggregation demonstrate superior performance, with Late Fusion consistently outperforming Blending across private and public datasets (compare results in Table 7 and Table 9 to those in Table 6 and Table 8).

**4. New downstream tasks.**

In addition to the downstream task outlined in Section 7: Conclusion and Future Work, we plan to further develop the multimodal dataset and release an updated version in 2025. The planned improvements include:

1. Extend the time range of the dataset to include data from 2023 and 2024, enabling longitudinal studies and trend analysis.
2. Incorporate new data sources, which will predominantly be utilized as additional input modalities to enhance the dataset's multimodal capabilities.
These will include:
  - Expanded communication data, such as new communication channels and enriched dialogue features (e.g., discussed topics), enabling research into the impact of communication on sales and financial activity.
  - Clickstream data from an internal site, covering visited pages, operations performed, and viewed advertisements, providing insights into customer online behavior.
3. Expanding Prediction Targets, which can serve as both input modalities and prediction targets. These will include:
  - Additional financial products (up to approximately 20), broadening the scope of target variables for prediction.
  - Customer Behavior Insights: customer churn, fraud detection, credit risk assessment
  - Segmentation of customer life cycles.

These updates will enhance the dataset's versatility for various tasks, enabling more comprehensive evaluations of representations. With a multi-task framework, we aim to facilitate innovative training methods, such as zero-shot learning, and support research in financial applications and customer behavior.

---

### Author Response · Authors · 2024-11-21

**5. Lack of novelty.**

We agree that our paper mostly concentrates on a new dataset and corresponding benchmark rather than developing innovative models. Our submission form chose the “datasets and benchmarks” as a Primary Area. As mentioned in Section 1. Introduction, we believe that papers in this area should mainly open new research questions and provide opportunities for future studies to develop new models rather than provide a strong innovative model and close the possibilities for significant improvements. As an example, we used an article about the HINT dataset (https://openreview.net/forum?id=kIPyTuEZuAK) that was accepted as a notable-top-25% paper at ICLR 2023.

We should emphasize that our dataset forces development of new techniques rather than simply applying existing models. Indeed, as detailed in Section 3.1 (Modalities), the event sequences in our dataset differ significantly from traditional time series due to their irregular temporal patterns and the inclusion of categorical features. For example, the methods suggested in the review ([1] Learning Deep Time-index Models for Time Series Forecasting; [2] Crossformer: Transformer Utilizing Cross-Dimension Dependency for Multivariate Time Series Forecasting) are primarily designed for predicting future values of a time series, which is fundamentally different from our downstream tasks, and heavily rely on temporal continuity. So they would require substantial adaptation to effectively handle the multiple data sources and asynchronous nature of our data.

Thus, we believe this contribution will drive meaningful advancements and help bridge a critical gap in research of multimodal sequential data analysis for real industrial use-cases.

We once again thank all the reviewers for assessing our work and for their valuable comments and advice, which helped us improve its presentation. We are happy to answer any additional questions you may have.

---

### Author Response · Authors · 2024-11-28

We sincerely thank all reviewers for their valuable comments and suggestions. Your insightful feedback has been instrumental in refining our work. We have revised the paper thoroughly to address your comments and improve its overall quality and clarity.

---

### Meta-Review · Area_Chair_m1Qd · 2024-12-20

**Metareview:**

# Summary and Recommendation for Rejection

---

## Strengths:
1. **Dataset Scale and Scope**:
   - The authors present the first large-scale, multimodal banking dataset (MBD) covering over 2 million corporate clients.
   - Data sources include bank transactions, geolocation data, customer support dialogues, and aggregated purchases.

2. **Benchmarking Contribution**:
   - Includes baseline evaluations comparing unimodal and multimodal methods, showcasing the advantages of multimodal approaches for financial tasks.

3. **Public Availability and Reproducibility**:
   - The dataset and experimental code are promised to be made publicly available, promoting transparency and fostering further research.

4. **Privacy Measures**:
   - Anonymization techniques are applied to protect user data while retaining significant information for downstream tasks.

---

## Weaknesses:
1. **Lack of Novelty**:
   - The paper primarily focuses on dataset creation and lacks significant methodological innovation.
   - For ICLR, novel methods tailored to the dataset are expected but are absent here.

2. **Sparse Details on Anonymization**:
   - While anonymization is discussed, there is insufficient quantitative evaluation of its impact on preserving temporal dependencies.
   - This raises concerns about data reliability, especially for fine-grained tasks like fraud detection.

3. **Limited Methodological Exploration**:
   - Advanced multimodal fusion techniques, such as cross-attention mechanisms, are not thoroughly explored.
   - Comparisons with state-of-the-art methods, including large language models (LLMs), are limited.

4. **Repetitive Modality Representations**:
   - The modalities are primarily tabular or text-based, limiting the diversity seen in standard multimodal datasets (e.g., including audio or video).

5. **Presentation Issues**:
   - The manuscript contains grammatical errors, unclear expressions, and inconsistent tense usage, detracting from readability.

---

## Authors' Mitigation:
1. **Advanced Fusion Techniques**:
   - The authors added preliminary results using early fusion techniques (e.g., CrossTransformer), but these showed no significant performance improvement.

2. **Expanded Benchmarking**:
   - Efforts to address label imbalance (e.g., stratified batching, oversampling) were included, but metric improvements were marginal.

3. **Anonymization Impact**:
   - Claims of minimal performance differences between anonymized and original datasets are made, but quantitative evaluations for complex tasks are missing.

4. **Future Dataset Enhancements**:
   - The authors promise to expand the dataset with new modalities and tasks in future versions but do not address current limitations.

---

## Remaining Weaknesses:
1. **Insufficient Novelty**:
   - The authors fail to provide novel frameworks or techniques tailored to the dataset.

2. **Unsubstantiated Claims**:
   - Assertions regarding anonymization and dataset reliability are not adequately supported by experiments or results.

3. **Limited Practicality**:
   - Metrics like AUC are used as proxies for campaign success, but their real-world implications for financial decision-making are not validated.

4. **Incomplete Benchmarks**:
   - Comparisons with modern multimodal learning approaches (e.g., MMBench) and integration with LLMs (e.g., GPT-4 or BloombergGPT) are insufficient.

---

## Decision Justification:
While the dataset's scale and intent are commendable, the paper falls short in several critical areas:

1. **Scope Misalignment**:
   - The focus on dataset release, with minimal methodological contributions, does not align with ICLR's emphasis on innovation.

2. **Reliability Concerns**:
   - The anonymized dataset's adequacy for sensitive tasks like fraud detection remains questionable.

3. **Limited Community Impact**:
   - Without novel methods or robust comparative analyses, the paper's contribution is constrained to the dataset.

4. **Quality Issues**:
   - Presentation flaws and lack of clarity weaken the submission.

---

## Recommendation:
**Reject**.   While the dataset provides a strong foundation for future research, significant revisions and expansions are necessary to meet the standards of a premier conference like ICLR.

**Additional Comments On Reviewer Discussion:**

Please refer to details in the above section.

---

### Decision · Program_Chairs · 2025-01-22

Reject